# Insights into the pathogenesis and differential diagnosis of clival lesions in an individual from a 16th-century-CE mass grave at Mohács (Southwestern Hungary)

Marcos De Andrés Montero[1,2]*, Luca Kis[1], Viktor Vig[1], Réka Kocsmár[1], Albert Zink[3], Frank Maixner[3], Bianca Mari[3], Alexandra Mussauer[3], Antal Sklánitz[4], András Palkó[5], Ádám Perényi[5,6], Árpád Szabó[7], Tímea Katalin Mai[7], Gábor Bertók[2], Olivier Dutour[8,9], Hélène Coqueugniot[8,9], György Pálfi[1]*, Olga Spekker[1,10]*

**1** Department of Biological Anthropology, University of Szeged, Szeged, Hungary, **2** Janus Pannonius Museum, Pécs, Hungary, **3** Institute for Mummy Studies, Eurac Research, Bolzano, Italy, **4** Central Quality Laboratory, Continental Automotive Hungary Ltd., Budapest, Hungary, **5** Department of Radiology, University of Szeged, Szeged, Hungary, **6** Department of Oto-Rhino-Laryngology and Head-Neck Surgery, University of Szeged, Szeged, Hungary, **7** Department of Forensic Medicine, University of Szeged, Szeged, Hungary, **8** École Pratique des Hautes Études, PSL University, Paris, France, **9** UMR (Archéosciences Bordeaux), Bordeaux Montaigne University, Pessac Cedex, France, **10** Ancient and Modern Human Genomics Research Group, University of Szeged, Szeged, Hungary

☯ These authors contributed equally to this work.
* olga.spekker@gmail.com (OS); mdeandres1993@gmail.com (MA); palfigy@bio.u-szeged.hu (GP)

## Abstract

The aim of our paper is to present and discuss in detail the pathological lesions suggestive of tuberculosis observed in a skeleton (**MMG3–75**) that was excavated from the 16th-century-CE mass grave No. 3 of the Mohács National Memorial Site (Sátorhely, southwestern Hungary). The differential diagnoses of the observed bony changes, with special attention to the clival alterations, are presented. During the macromorphological, radiological, and digital microscopic examination of **MMG3–75**, the skull base showed mild cortical erosion and multiple, well-circumscribed osteolytic lesions at the clivus clearly evidenced by 3D imaging. In addition, endocranial granular impressions and abnormal blood vessel impressions were observed in multiple locations on the inner skull surface. Based on the differential diagnosis of the clival changes and their co-occurrence with endocranial alterations indicative of tuberculous meningitis (granular impressions and abnormal blood vessel impressions), they were most likely due to tuberculous involvement of the skull base. Additional aDNA analysis provided no evidence for the presence of *Mycobacterium tuberculosis* DNA in **MMG3–75**. To the best of our knowledge, **MMG3–75** is the first reported archaeological case of tuberculous clival osteomyelitis with associated meningitis, giving us a unique insight into the occurrence of an extremely rare manifestation of tuberculosis in mediaeval Hungary.

**Data availability statement:** All relevant data are within the manuscript and its Supporting Information files.

**Funding:** This study was carried out within the framework of the Mohács 500 project, with the support of the Ministry of Agriculture of the Hungarian Government (government decree nos. 1159/2023., IV. 27. and 1288/2024., IX. 19.), awarded to GP. This work was also funded by the University Research Scholarship Programme – Cooperative Doctoral Programme 2024 (2024-2.1.2-EKÖP-KDP) of the Hungarian Ministry of Culture and Innovation (grant agreement no. EKÖP-KDP-24-SZTE-2) to MAM, by the University of Szeged Open Access Fund (grant agreement no. 7926) to OS, and by the STARTING_25 sub-programme of the National Research Excellence Programme, under the management of the Hungarian National Research, Development and Innovation Office (grant agreement no. STARTING 153194) to OS. Moreover, this research was supported by the Competence Centre of the Life Sciences Cluster of the Centre of Excellence for Interdisciplinary Research, Development and Innovation of the University of Szeged. This project received funding also from the European Research Council (ERC) under the European Union's Horizon 2020 research and innovation programme (grant agreement no. 856453 ERC-2019-SyG). The funders had no role in the study design; in the collection, analysis, and interpretation of data; in the writing of the manuscript; and in the decision to submit the manuscript for publication.

**Competing interests:** The authors have declared that no competing interests exist.

## Introduction

Tuberculosis (TB) is a bacterial infection that is caused by a group of genetically closely related mycobacterial species (e.g., *Mycobacterium tuberculosis* or *Mycobacterium africanum*), collectively known as the *Mycobacterium tuberculosis* complex (MTBC) [1–6]. Despite about 100 years of vaccination with the Bacillus Calmette–Guérin (BCG) vaccine and about 80 years of anti-TB chemotherapy with antibiotics (e.g., isoniazid, rifampicin or streptomycin), TB still affects a significant proportion of the world's population [7–11]. According to the latest estimates of the World Health Organization (WHO) [12], about a quarter of the world's current population (~1.8 billion people) is infected with TB bacilli, but most of these people will not go on to develop clinically active TB disease and are likely to have latent TB infection for the rest of their lives [6,7,9,10,13]. In 2023, about 10.8 million people contracted TB and about 1.25 million patients died of the disease worldwide [12] – making TB the world's leading cause of death from a single infectious agent again (replacing COVID-19) [5,9,11,12].

TB bacilli are mainly spread from person to person through the air; this is why the disease primarily affects the lungs (i.e., pulmonary TB), although other organs or tissues of the human body may become involved (i.e., extra-pulmonary TB) [1,2,4,13]. Nowadays, extra-pulmonary TB (EPTB) accounts for about one fifth of all cases of clinically active TB disease, and usually results from haematogenous or lymphogenous spread of TB bacilli and subsequent involvement of one or multiple non-pulmonary sites (e.g., pleura, gastro-intestinal tract or central nervous system) [2,14–17]. Skeletal or osteoarticular TB (OATB), which refers to a rare form of TB that affects the bones and/or joints, accounts for about 10% of all cases of EPTB today [15,18,19]. Any bone or joint can become involved by TB but the spine (~50%) and large weight-bearing joints (e.g., hip and knee; ~30%) are the most common skeletal sites of the disease due to their rich vascular supply [14,18–22].

Tuberculosis of the skull, involving one or multiple cranial bones, is a very rare clinical entity, accounting for up to 1% of all cases of OATB [23–28]. Although the prevailing view is that TB bacilli can enter the cancellous bone of the cranial bones by haematogenous spread, some researchers suggest that lymphatic vessels may be a more likely route for the pathogens to reach the skull [29–37]. This hypothesis would explain the rare occurrence of cranial TB as the skull has a poor lymphatic supply that hinders the lymphogenous spread of TB bacilli from the primary extra-cranial site to the cranial bones [30,32,34,35,37–39]. In addition to haematogenous and lymphogenous spread, TB of the skull may be secondary to contiguous extension of the infection from adjacent structures (e.g., meninges or vertebrae) or direct traumatic or surgical inoculation of TB bacilli into the cranial bones [30,31,35,36,40].

Infiltration of TB bacilli into the cancellous bone of the skull causes granulomatous inflammation, leading to the development of tubercles in the red bone marrow [33,35,41–43]. As these tubercles gradually expend and coalesce while undergoing caseous necrosis, growth of the original intra-osseous abscess(es) occurs along with formation of additional intra-osseous abscesses within the infected cancellous bone [35,41,43]. In addition, there is a reduction of the blood supply to the cancellous bone

due to TB involvement and consequent obliteration of the blood vessels [30,35,44,45]. Any or all of the aforementioned pathological processes lead to necrosis and subsequent resorption of the bone trabeculae, ultimately resulting in the development of single or coalescing, spherical or ovoid osteolytic lesions within the cancellous bone, filled with TB granulation tissue [29,30,35,39,41–44]. As the pathological process progresses, the TB infection may spread from the cancellous bone to the cortical layers of the affected cranial bone(s), leading to perforation(s) [29–30]. In the end, the entire thickness of the affected cranial bone(s) may be destroyed [29,30,32,41,46,47]. At the perforation site(s), an accumulation of TB granulation tissue develops, which can cause cortical erosion of the adjacent bone surfaces [28,30,35,40,44,48–51].

Although TB osteomyelitis can develop in any bone of the skull vault (i.e., TB calvarial osteomyelitis) or skull base (i.e., TB skull-base osteomyelitis), the frontal and parietal bones are the most common sites of involvement [26,29,31,35,36,45,47]. This is most likely due to the fact that these cranial bones contain a greater amount of cancellous bone; and therefore, highly vascularised red bone marrow [26,29,35,36,45]. TB osteomyelitis of the skull base is extremely rare with only a few cases reported in the modern medical literature [21,23,24,27,52]. It can result either from the haematogenous spread of TB bacilli from a primary focus elsewhere in the host's body (e.g., the lungs) or from the direct extension of the TB infection from an adjacent site (e.g., the sphenoid sinus or cranio-vertebral junction/CVJ) [21,23,52,53].

The clivus (of Blumenbach) is a midline, downward-sloping bony structure at the central skull base (posterior to the *dorsum sellae* and anterior to the *foramen magnum*), which is formed by the fusion of the posterior part of the sphenoid body (basisphenoid) and the basilar part of the occipital bone (basiocciput) at the spheno-occipital synchondrosis [53–58]. Tuberculous involvement of the clivus usually results from contiguous spread of the infection from the CVJ [21,59–60], while TB clival osteomyelitis sparing the CVJ is rare [50,59,61–66]. It has been suggested that the clivus may also become infected by TB bacilli via alternative routes, such as haematogenous dissemination or extension from adjacent pharyngeal lymph nodes [23,50,53,59,61,65–67]. The disease may present with extensive cortical erosion and osteolytic lesions accompanied by accumulation of TB granulation tissue in the area of the clivus [50,51,53,58,68,69]; in TB osteomyelitis, the osteolytic lesions are typically well-defined, with minimal or no marginal sclerosis and little or no reactive new bone formation in the surrounding bone, while sequestration is relatively uncommon [70–73]. Associated TB meningitis (TBM) can further complicate TB clival osteomyelitis [21,23,27,28,52,53,65,67].

Archaeological cases representing rare forms of TB, such as TB clival osteomyelitis, are of particular scientific significance, as they can expand our knowledge and understanding of how the disease manifested, progressed, and spread across space and time in past human populations. Detailed presentation and discussion of such cases can serve as valuable references in the palaeopathological evaluation of skeletal assemblages, since they facilitate the identification and interpretation of similar instances, improve diagnostic accuracy, and broaden the spectrum of TB forms recognisable in human osteoarchaeological series. Furthermore, these cases also hold relevance for modern medical practice, as they provide important comparative data that may assist clinicians in diagnosing TB manifestations that, owing to their rarity today, are almost entirely unfamiliar (e.g., TB clival osteomyelitis). Building on these considerations, the aim of our paper is to present and discuss in detail a younger adult male (**MMG3–75**) from the 16th-century-CE mass grave No. 3 of the Mohács National Memorial Site (Sátorhely, southwestern Hungary), who represents a unique manifestation of TB with concomitant involvement of the skull base (clivus) and the meninges. To the best of our knowledge, **MMG3–75** is the first reported archaeological case of TB clival osteomyelitis with associated meningitis.

## Materials and methods

### Material

The Mohács National Memorial Site is located to the east of the present-day village of Sátorhely (Baranya county, southwestern Hungary) (Fig 1a) [74–76]. Between 2020 and 2022, a team of archaeologists from the Janus Pannonius Museum (Pécs, Hungary) and anthropologists from the Department of Biological Anthropology, University of Szeged

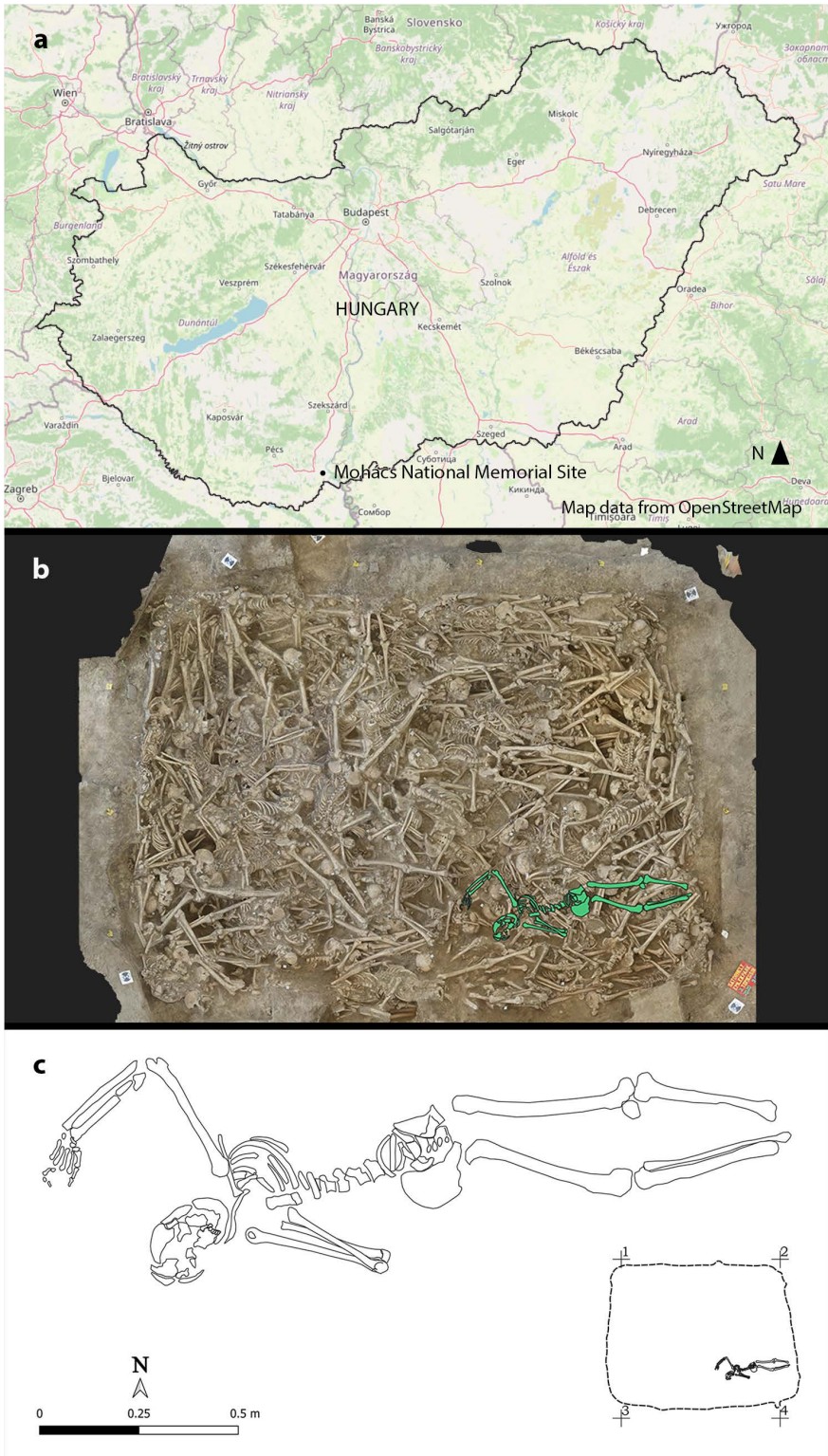

**Fig 1. a)** Map of Hungary showing the location of mass grave No. 3 of the Mohács National Memorial Site (contains information from Open-StreetMap and OpenStreetMap Foundation, which is made available under the Open Database License (**https://www.openstreetmap.org/copyright**)); **b)** Orthophoto of mass grave No. 3 of the Mohács National Memorial Site showing the location of the skeleton of MMG3-75 (orthophoto

by Gábor Bertók and Béla Simon; source of the orthophoto: Photo Collection and Archaeological Database of the Janus Pannonius Museum, Pécs, Hungary); and c) Drawing of the skeleton of MMG3-75 and its location in mass grave No. 3 of the Mohács National Memorial Site (drawing by Zsófia Simon and Gábor Nagy; source of the drawing: Photo Collection and Archaeological Database of the Janus Pannonius Museum, Pécs, Hungary).

(Szeged, Hungary) excavated one of the five known mass graves at the site [74,75,77,78]. The burial pit known as mass grave No. 3 had an irregular rectangular shape with dimensions of: 4.0 metres long on the northern side, 4.8 metres long on the southern side, 3.25 metres wide, a depth of about 1.4 metres, and a total surface area of about 15 square metres (Fig 1b) [74,77,79]. This pit contained the partially commingled bone remains of approximately 320 individuals (based on a preliminary count of right femora; a more accurate count is still being finalised) [78]. They were buried in the aftermath of the Battle of Mohács, which took place on 29th August 1526 [74,78,79]. This battle is considered to be a decisive event in the history of Hungary as it marked the end of more than 150 years of the Hungarian resistance to the various attempts of Ottoman conquest [80]. It is important to note that, to date, several hypotheses have been proposed regarding the identity of the individuals buried in the five known mass graves at the Mohács National Memorial Site, suggesting that they may have been soldiers, members of the Hungarian camp or civilians from nearby settlements [77,79]. Ongoing research seeks to reassess these hypotheses and to shed further light on both the identities of these individuals and the circumstances of their death – whether they perished in the battle, died while retreating or were executed afterwards. Among the skeletons found in mass grave No. 3, **MMG3–75**, lying prone, was excavated from the southeastern corner of the pit (Fig 1b, 1c); no grave goods could be associated with this individual. The bone remains of **MMG3–75** are housed in the facilities of the Department of Biological Anthropology, University of Szeged (Szeged, Hungary).

### Ethics statement

Specimen number: **MMG3–75** (archaeological site: mass grave No. 3, Mohács National Memorial Site (Sátorhely, Baranya county, southwestern Hungary); inventory no. COM.S-75).

The human skeleton (**MMG3–75**), which was evaluated in the described study, is housed in the Department of Biological Anthropology, University of Szeged, in Szeged, Hungary. Access to the specimen was granted by the Department of Biological Anthropology, University of Szeged (Közép fasor 52, H-6726 Szeged, Hungary).

No permits were required for the described study, which complied with all relevant regulations. The research has been conducted in an ethically responsible manner – the bone remains of **MMG3–75** have been examined with dignity and respect.

### Methods

**Macromorphological investigation.** Prior to the palaeopathological examination of the bone remains of **MMG3–75**, a basic anthropological analysis was carried out on the fairly preserved and almost complete skeleton (Fig 2) with the aim of reconstructing the biological profile of this individual. Age at death was estimated (Todd [81], Nemeskéri & colleagues [82], Brooks & Suchey [83], Buckberry & Chamberlain [84], and Schmitt [85]) and biological sex was determined (Éry & co-workers [86], Brůžek [87], and Brůžek & colleagues [88]) using macromorphological and/or metric methods that are among the standards of practice in the field of bioarchaeology. Based on the results of these investigations, the skeleton of **MMG3–75** presented morphological features and metric values consistent with a male who likely died between the ages of 30 and 39. The palaeopathological assessment following the basic anthropological analysis was carried out with the naked eye. It focused on the detection of skeletal lesions (Table 1) that have been associated with different forms of TB (pulmonary TB/TB pleurisy [e.g., 89–97], TBM [e.g., 98–107],

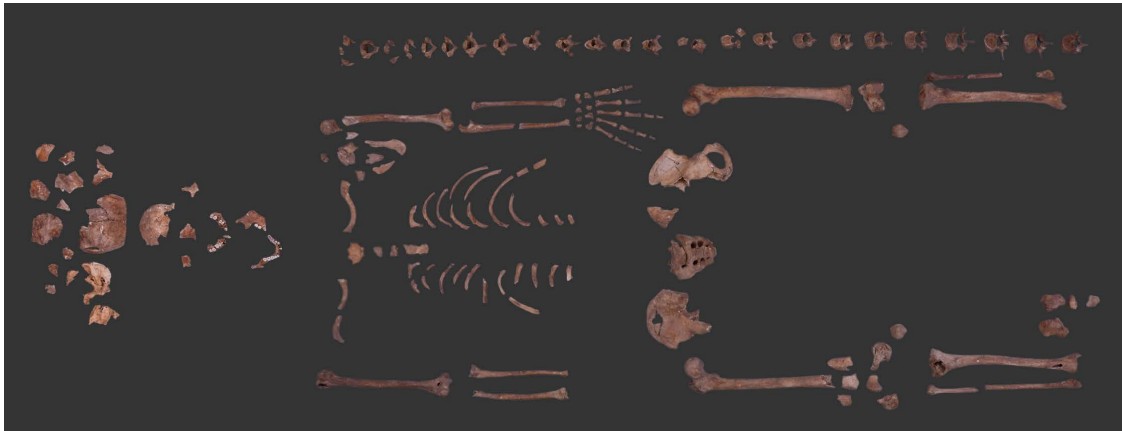

**Fig 2. State-of-preservation of the skeleton of MMG3-75 (photo by Marcos De Andrés Montero, Luca Kis, and Olga Spekker).**

**Table 1. List of diagnostic criteria for TB that were considered during the palaeopathological evaluation of the skeleton of MMG3-75.**

| Disease type | Diagnostic criteria |
|---|---|
| **Pulmonary tuberculosis/ tuberculous pleurisy** [e.g., 89–97] | Subperiosteal new bone formations and/or erosive changes on the ribs (predominantly on the visceral costal surfaces), and signs of diffuse, bilateral, symmetrical periostitis on the diaphysis of the short and/or long tubular bones (i.e., hypertrophic pulmonary osteopathy) |
| **Tuberculous meningitis** [e.g., 98–107] | Endocranial abnormally pronounced digital impressions (APDIs), endocranial abnormal blood vessel impressions (ABVIs), endocranial periosteal appositions (PAs), and endocranial granular impressions (GIs) |
| **Different forms of osteoarticular tuberculosis** [e.g., 108–113] | **Spinal tuberculosis** (i.e., the combination of tuberculous vertebral osteomyelitis and arthritis): signs of hypervascularisation on the anterior and/or lateral aspects of one or multiple vertebral bodies – in the form of multiple, circumferential, smooth-walled, resorptive pits (enlarged vascular foramina); osteolytic lesions and/or erosive changes on one or multiple vertebral bodies and/or posterior elements; destruction, collapse, and/or fusion of one or multiple vertebral bodies and/or posterior elements; cortical remodelling and/or reactive new bone formations on one or multiple vertebral surfaces; destruction, subluxation, dislocation, and/or ankylosis of one or multiple intervertebral joints; and signs of a cold abscess in the pelvic area (e.g., hip bones and femora) – in the form of erosive changes, cortical remodelling, and/or reactive new bone formations |
| | **Extra-spinal tuberculous osteomyelitis**: osteolytic lesions and/or erosive changes on one or multiple extra-spinal bones; and cortical remodelling and/or reactive new bone formations on the surface(s) of one or multiple extra-spinal bones |
| | **Extra-spinal tuberculous arthritis**: peri-articular osteoporosis, osteolytic lesions, and/or erosive changes in the subchondral bone of one or multiple extra-spinal joints; "kissing sequestra" on the opposing sides of one or multiple extra-spinal joints; and subluxation, dislocation, and/or ankylosis of one or multiple extra-spinal joints |

and OATB [e.g., 108–113]) in the palaeopathological literature. In addition to the aforementioned diagnostic criteria for TB, stress indicators – such as *cribra orbitalia* and *cribra cranii* – were also considered in the course of the palaeopathological study of **MMG3–75** as these alterations have been frequently recorded in cases of TB [114–116]. Although the overall state-of-preservation of the skeleton of **MMG3–75** was fairly good and almost complete (Fig 2), the taphonomic damage and the missing elements – particularly in skeletal regions regarded as predilection sites for TB-related lesions (e.g., vertebrae; ribs; large, weight-bearing joints; and the inner surface of the skull) significantly hampered the macromorphological observations. For instance, while most of the vertebrae and ribs were present, the latter, in particular, were extremely fragmentary and incomplete. The skull was considerably fragmentary, with some missing parts and several *post-mortem* damaged areas on the endocranial surface. Some of the large, weight-bearing joints also showed clear *post-mortem* damage (Fig 2).

**Radiological and digital microscopic examinations, and 3D imaging.** The *post-mortem* fragmented bones of the skull base of **MMG3–75** were not reconstructed for the radiological and digital microscopic examination. Only the fracture line running along the right side of the lower part of the clivus and extending into the *foramen magnum* was glued in order to allow the clivus and the petrous parts of the temporal bones to be seen together.

Cone beam computed tomographic (CBCT) analysis was performed on some skull base fragments of **MMG3–75** at the Király Dental X-ray and CBCT Centre (Szeged, Hungary). The CBCT images were acquired using a NewTom VGi EVO unit with a resolution of 150 µm. The CBCT slices on DICOM format were open on TIVMI software [117] to perform 3D reconstruction. A manual segmentation of the largest cysts, after process by greyscale thresholding, was performed on each slice. A 3D reconstruction was obtained upon a surface extraction.

The same skull fragments from **MMG3–75** were subjected to digital microscopic analysis by the Central Quality Laboratory, Continental Automotive Hungary Ltd. (Budapest, Hungary). The endocranial surface of the selected skull fragments of **MMG3–75** was scanned and analysed using a Keyence VHX-5000 digital microscope.

## Palaeogenetic analysis

Hard tissue samples (the upper right lateral incisor and a fragment of the left parietal bone) of **MMG3–75** were subjected to DNA-based analysis as described by Jäger & colleagues [118]. In brief, DNA was extracted from 68 mg (tooth) and 87 mg (skull fragment) samples using a linear polyacrylamide-based extraction technique [119]. DNA extracts were converted into double-indexed and double-stranded Illumina libraries following the protocol by Meyer & Kircher [120] and Kircher & co-workers [121], and subsequently sequenced on the Illumina HiSeq X platform using the 151 bp paired-end sequencing kit. The shotgun datasets were screened for traces of *M. tuberculosis* DNA using MetaPhlan v4.1.1 [122], as well as mapping against the *M. tuberculosis* H37Rv reference genome (NC_000962.3) [123] and MTB marker genes according to the bioinformatics pipeline described by Jäger & colleagues [118] with minor modifications. Human DNA detection and analysis in the datasets was performed following the bioinformatics pipeline outlined by Laffranchi & co-workers (see S1 Text in [124]).

## Results

### Macromorphological investigation

Based on the diagnostic criteria compiled from the palaeopathological literature and summarised in Table 1, the skull of **MMG3–75** showed numerous bony changes that are suggestive of TB.

The endocranial surface of the upper-middle clivus exhibited mild cortical erosion accompanied by multiple, very small (~1–3 mm in diameter), well-circumscribed perforating osteolytic lesions (Figs 3 and 4). In addition to the above, two types of endocranial alterations – granular impressions (GIs) and abnormal blood vessel impressions (ABVIs) – were noted on the inner surface of the skull of **MMG3–75**, despite the *post-mortem* absence and damage of some areas. Multifocal GIs were observed on both sides of the frontal bone, at the two junctions of the squamous part with the orbital parts (Fig 5a, 5d–5g–5g), on the right greater wing of the sphenoid bone (Fig 6a), and on the squamous part of both temporal bones, particularly close to the sphenosquamosal suture (Fig 6b–6d). Unifocal GIs were registered on the right orbital part of the frontal bone, close to the ethmoidal notch (Fig 5b–5c), on the right parietal bone, close to the posterior end of the squamous suture (Fig 6f–6g), on the left greater wing of the sphenoid bone (Fig 6e), and on the occipital bone, in the right cerebral fossa (Fig 6h). The observed GIs covered only a very small proportion of the inner skull surface in most of the affected areas. Besides GIs, ABVIs were recorded on the frontal (Fig 7a), both parietal (Fig 7b–c), and the occipital (Fig 6h) bones, scattered all over the endocranial surface.

It should be noted that no lesions indicative of TB were observed in the postcranial skeleton of **MMG3–75**, based on the diagnostic criteria compiled from the palaeopathological literature (summarised in Table 1).

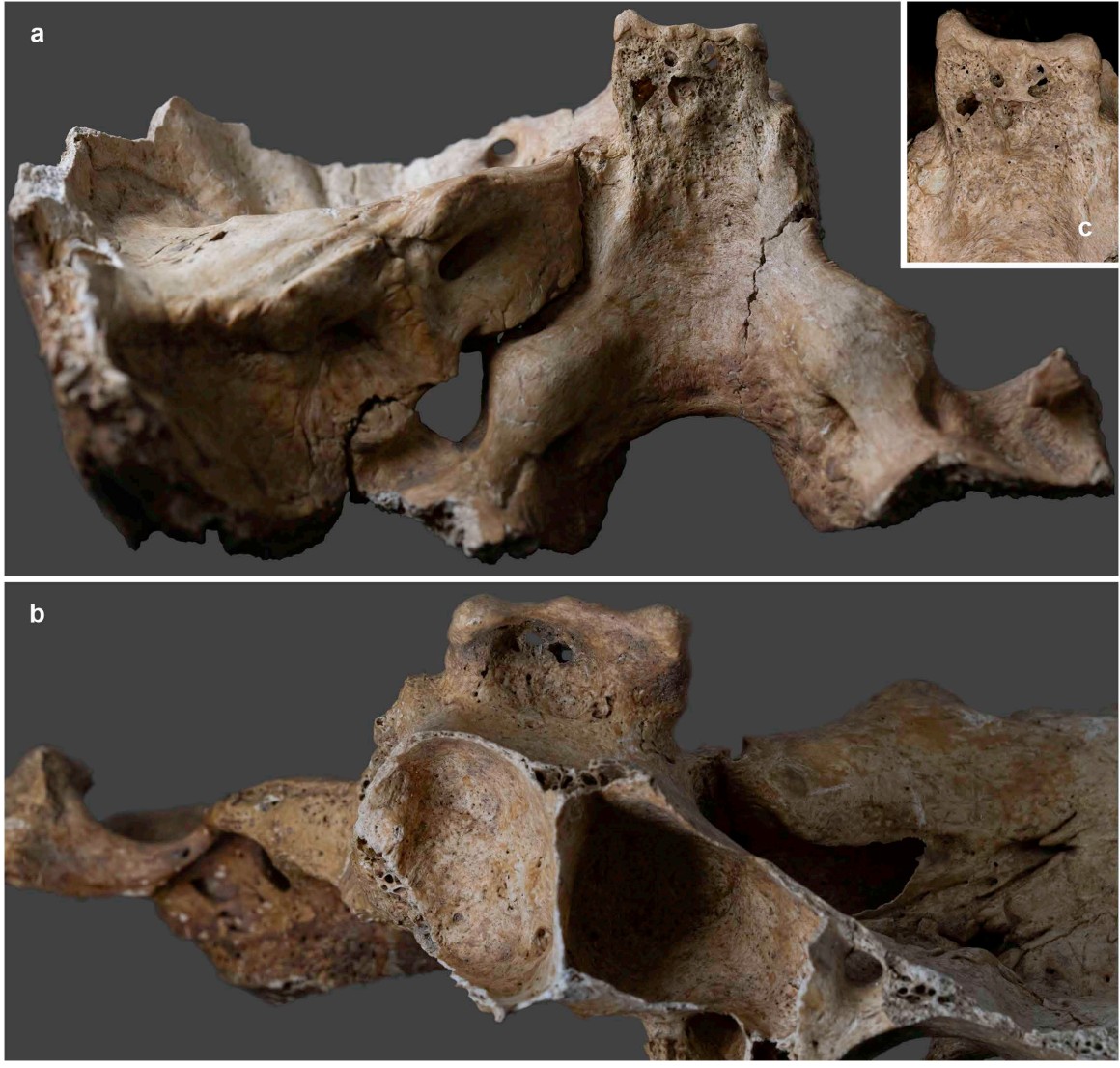

**Fig 3. Mild cortical erosion accompanied by multiple, very small, well-circumscribed perforating osteolytic lesions on the endocranial surface of the upper-middle clivus of MMG3-75 (photos by Marcos De Andrés Montero, Luca Kis, and Olga Spekker): a) posterior view and b) anterior view. c) Digital microscopic image of the mild cortical erosion and osteolytic lesions on the endocranial surface of the upper-middle clivus of MMG3-75 (close-up) (image by Antal Sklánitz).**

### Radiological and digital microscopic examinations, and 3D imaging

Both middle ears, both petrous apexes, and the sphenoid sinus are well-developed and well-pneumatised, and their bony structures and bony margins are intact (Fig 8b). There is no evidence of inflammatory skeletal lesions. On both sides, the inner ear is anatomically intact, and the bony canals of the facial nerve and internal carotid artery are normal (Fig 8b). In the upper, middle, and lower parts of the clivus, three thin-walled, hypodense, larger (8 mm, 5 mm, and 5 mm in diameter) bone cysts, without thick sclerotic margins or bone sequestration, are observed (Fig 8a). Three-dimensional reconstruction (Fig 9) enabled the spatial visualisation of these three larger bone cysts, which were clearly distinguishable within the medullary space, among smaller myelitic changes. On the dorsal surface of the upper-middle clivus, the cortical bone is fragmented and incomplete (Figs 3c and 8c).

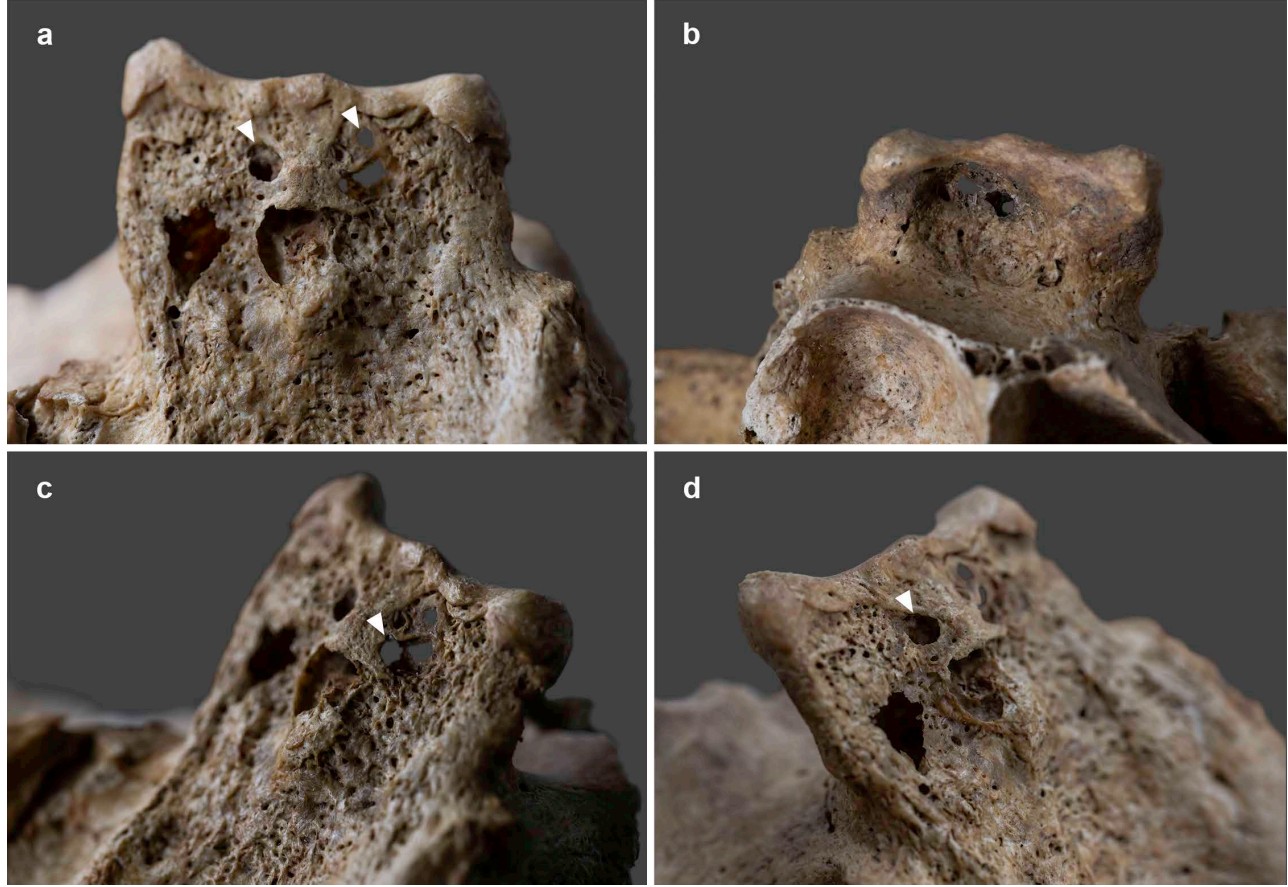

**Fig 4. Close-up of the mild cortical erosion accompanied by multiple, very small, well-circumscribed perforating osteolytic lesions (white arrows) on the endocranial surface of the upper-middle clivus of MMG3-75 (photos by Marcos De Andrés Montero, Luca Kis, and Olga Spek-ker): a) posterior view, b) anterior view, c) right postero-lateral view, and d) left postero-lateral view.**

## Palaeogenetic analysis

The DNA-based analysis revealed an overall poor endogenous DNA quality in the analysed hard tissue samples (a tooth and a skull fragment) of **MMG3–75**. The endogenous human DNA content was found to be less than 0.01%, however, sequences mapping to the human reference genome (build hg19) exhibited a damage pattern typical for ancient DNA (aDNA) (see S1 Fig). By combining the data from both hard tissue samples, approximately 3,000 human sequences were recovered, enabling genetic sex determination using the method described by Mittnik & colleagues [125]. This analysis identified a genetic sex consistent with XY (male), but not XX (female), for **MMG3–75**, which aligns with the morphological findings. No further information could be derived from the analysis of the mitochondrial DNA (mtDNA) sequences. Additional microbial screening revealed no traces of *M. tuberculosis* DNA. The majority of the non-human DNA could be assigned to different bacterial species isolated from various soil environments. Overall, both samples display poor DNA preservation not allowing further molecular characterisation of the individual.

## Discussion

The pathological skeletal lesions observed in the skull remains of **MMG3–75** correspond to several diagnostic criteria for TB compiled from the palaeopathological literature (summarised in Table 1) suggesting that this younger adult male may

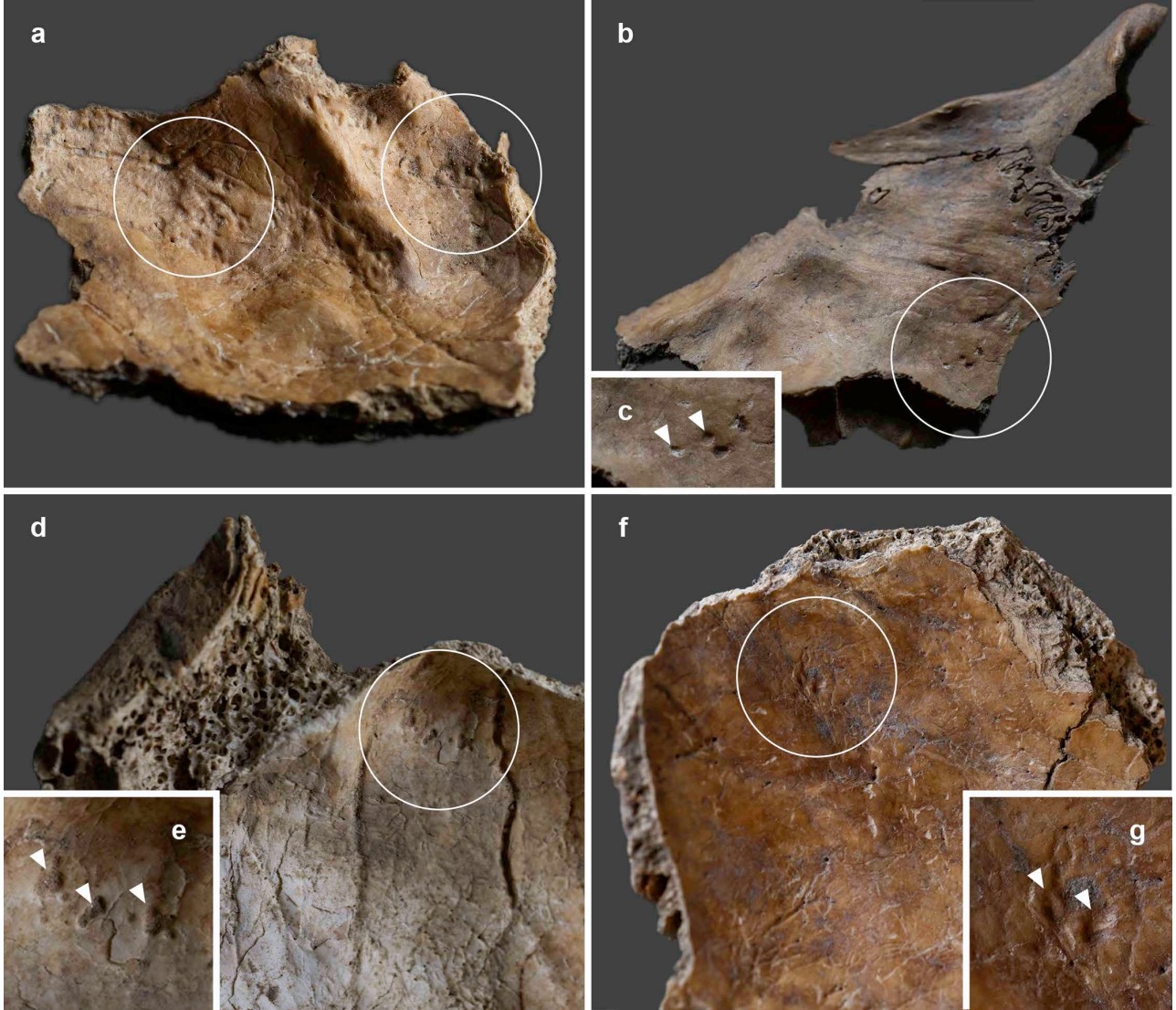

**Fig 5. Endocranial granular impressions (white circles and arrows) a)** at the junction of the squamous and right orbital parts of the frontal bone of MMG3-75, **b)** on the right orbital part of the frontal bone of MMG3-75, **c)** on the right orbital part of the frontal bone (close to the ethmoid notch) of MMG3-75 (close-up), **d)** at the junction of the squamous and left orbital parts of the frontal bone of MMG3-75, **e)** at the junction of the squamous and left orbital parts of the frontal bone of MMG3-75 (close-up), **f)** on the squamous part of the frontal bone (on the right side, close to the coronal suture) of MMG3-75, and **g)** on the squamous part of the frontal bone (on the right side, close to the coronal suture) of MMG3-75 (close-up). (Photos by Marcos De Andrés Montero, Luca Kis, and Olga Spekker.).

have suffered from TB with concomitant involvement of the skull base (clivus) and the meninges. To substantiate this hypothesis, we employed a three-step analytical approach: (1) we attempted to reconstruct the possible pathogenetic scenarios that could account for the development of the cranial bony changes observed in **MMG3–75**, assuming that TB was indeed the underlying cause, thereby demonstrating that the formation of such lesions is plausible in the context of the disease; (2) we sought to provide corroborative evidence that **MMG3–75** indeed suffered from TB at the time of death, thereby reinforcing that it is plausible for the disease to have been the underlying cause of the detected cranial alterations; and (3) we performed a detailed differential diagnosis of the clival lesions observed in **MMG3–75**, thereby taking

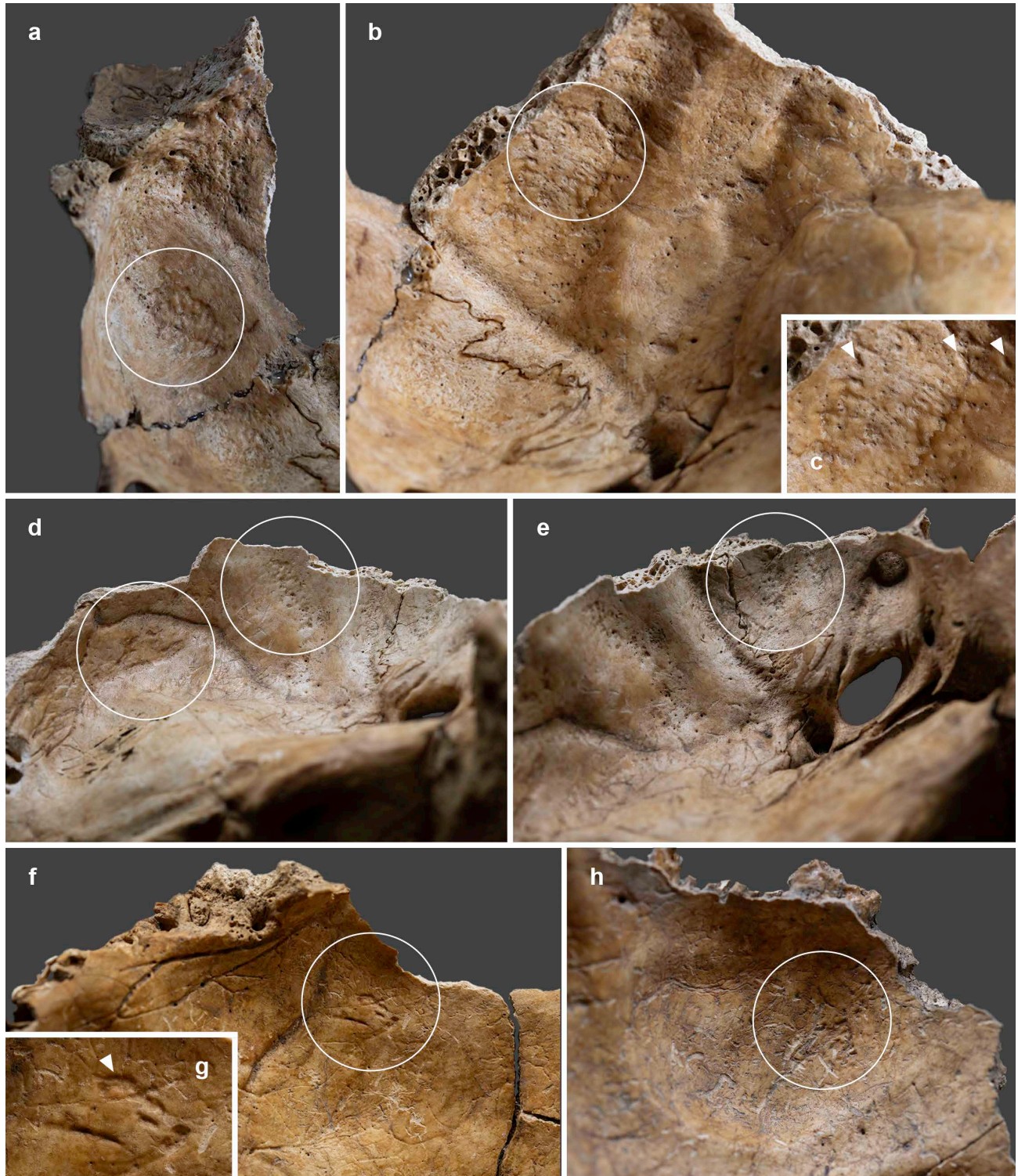

**Fig 6. Endocranial granular impressions (white circles and arrows) on the a) right greater wing of the sphenoid bone of MMG3-75, b) squa-mous part of the right temporal bone (close to the sphenosquamosal suture) of MMG3-75, c) squamous part of the right temporal bone (close to the sphenosquamosal suture) of MMG3-75 (close-up), d) squamous part of the left temporal bone (at the middle and close to the sphe-nosquamosal suture) of MMG3-75, e) left greater wing of the sphenoid bone of MMG3-75, f) right parietal bone (close to the posterior end of**

the squamous suture) of MMG3-75, g) right parietal bone (close to the posterior end of the squamous suture) of MMG3-75 (close-up), and h) occipital bone (right cerebral fossa) of MMG3-75. (Photos by Marcos De Andrés Montero, Luca Kis, and Olga Spekker.).

into account the possibility that another pathological condition – concurrently present with TB in this younger adult male – could have contributed to the development of these bony changes.

### Pathogenesis of TB in MMG3–75

Assuming our hypothesis (i.e., **MMG3–75** suffered from TB at the time of death, and that this disease caused the observed cranial bony changes) is correct, two plausible pathogenetic scenarios can be proposed: (1) TB first affected the clivus (i.e., TB clival osteomyelitis) and subsequently extended to the meninges (i.e., TBM) (S2 Fig), or (2) it involved the meninges first (i.e., TBM) and then reached the clivus (i.e., TB clival osteomyelitis) (S3 Fig).

1) Although TB clival osteomyelitis usually results from contiguous spread of the TB infection from adjacent structures, predominantly the CVJ [21,59,60], there was no skeletal evidence of TB involvement of the atlanto-occipital joint, sphenoid sinus, or either petrous pyramids in **MMG3-75**, making this scenario less likely. Rather, the disease may have developed following haematogenous spread of TB bacilli from a primary focus elsewhere in the body of **MMG3-75** [50,59,65–67] that unfortunately could not be identified during the palaeopathological examination of the skeleton. In **MMG3-75**, the presence of TB bacilli within the cancellous bone of the clivus may have triggered granulomatous inflammation with subsequent formation of the three larger, well-circumscribed osteolytic lesions observed within the upper, middle, and lower parts of the clivus (Figs 8a and 9) [29,30,33,35,39,41–44]. As the pathological process progressed, the TB infection may have spread from the cancellous bone to the cortical layers, resulting in perforations [29–30], with the consequent development of the multiple, very small, well-circumscribed osteolytic lesions observed in the upper-middle clivus of **MMG3-75** (Figs 3–4). On the endocranial surface, the mild cortical erosion of the upper-middle clivus suggests the presence of an accumulation of TB granulation tissue at the sites of perforation (Figs 3–4 and 8c) [28,30,35,40,44,48–51].

   In addition to the above, the presence of endocranial alterations indicative of TBM (GIs and ABVIs) in multiple locations on the inner skull surface of **MMG3–75** (Figs 5–7) suggests that the TB infection may have extended from the clivus towards the meninges, at least the *dura mater encephali*. In the palaeopathological literature, endocranial GIs have been described as specific signs of TBM [99–101,103,104,107]. Their development is secondary to the pressure atrophy of the cortical bone layers, caused by the tubercles formed on the *dura mater encephali* as a result of its TB granulomatous inflammation [99,103,104,107]. Unlike GIs, endocranial ABVIs are not specific to TBM as they can be caused by many other aetiologies (e.g., bacterial meningitis, traumas, scurvy, brain tumours, and haemorrhages) [98,100–102,105,107,126,127]. In TBM, their formation is secondary to the healing of an epidural haematoma and associated vascularisation that develops following granulomatous inflammation and consequent disruption of at least some of the meningeal blood vessels and/or dural venous sinuses [98,100,101,105,107]; and

2) An alternative scenario is that TB clival osteomyelitis may have developed following contiguous spread of the TB infection from the meninges towards the clivus of **MMG3-75**. In TBM, the primary event is the formation of a thick, gelatinous inflammatory exudate between the two layers of the leptomeninges (i.e., the *pia mater encephali* and the *arachnoid mater encephali*) [52,128–131]. The TB meningeal exudate is initially confined to the basal parts of the brain (the superior aspect of the cerebellum, the floor of the third ventricle, the anteromedial surface of the temporal lobes, and the inferomedial surface of the frontal lobes) but can rapidly spread to the basal cisterns (i.e., the interpeduncular and chiasmatic (suprasellar) cisterns, the former of which is adjacent to the upper-middle clivus) [52,128–130,132,133].

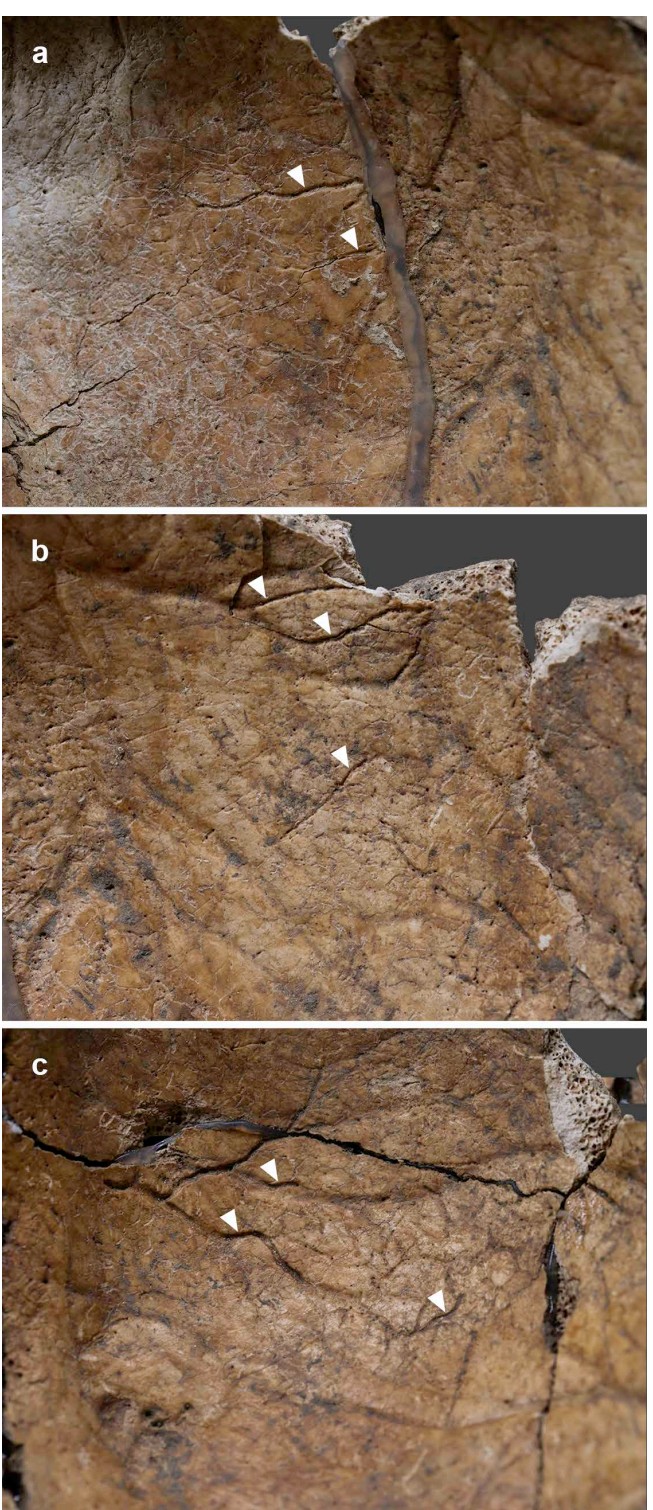

**Fig 7. Endocranial abnormal blood vessel impressions (white arrows) on the a) squamous part of the frontal bone (on the left side, close to the coronal suture) of MMG3-75, b) left parietal bone (at the middle) of MMG3-75, and c) right parietal bone (close to the sagittal suture) of MMG3-75. (Photos by Marcos De Andrés Montero, Luca Kis, and Olga Spekker.).**

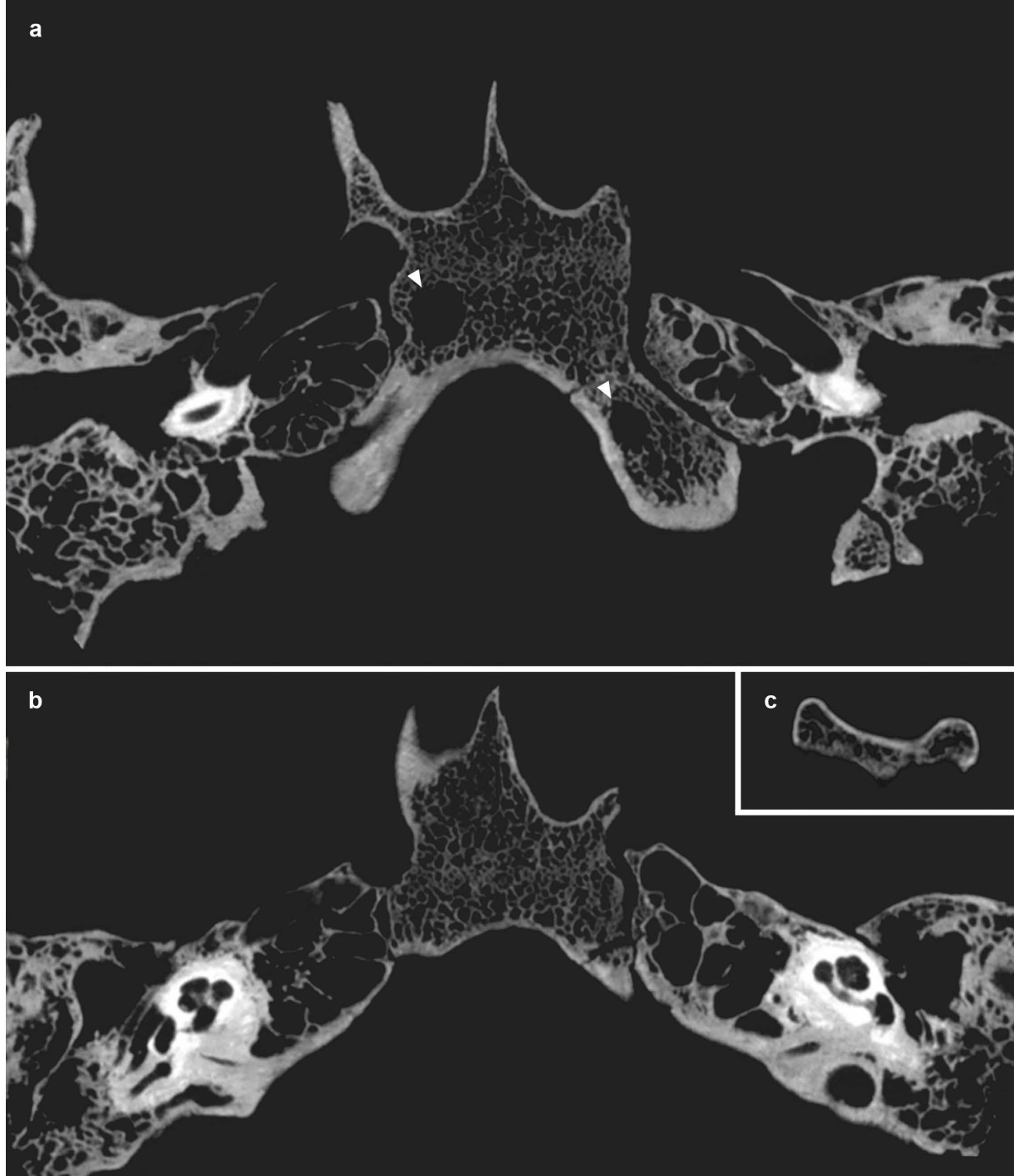

**Fig 8. Cone-beam computed tomographic scans of some skull base fragments of MMG3-75 (clivus and petrous parts of the temporal bones; slice thickness: 150 μm) (images by Árpád Szabó and Ádám Perényi): a) two larger thin-walled, hypodense bone cysts (8 mm and 5 mm in diameter), without thick sclerotic margins or bone sequestration (white arrows) in the clivus, b) anatomically intact middle and inner ears, and petrous apexes, and c) fragmented and incomplete cortical bone on the dorsal surface of the clivus.**

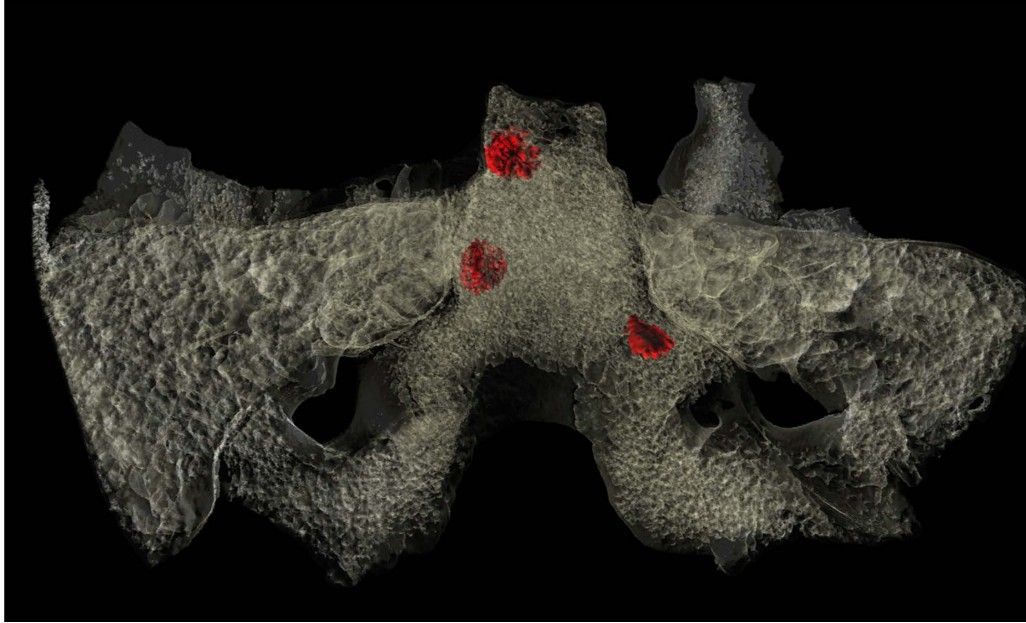

**Fig 9. Three-dimensional imaging of the clival lesions of MMG3-75: reconstruction of three larger intra-cancellous cystic cavities (highlighted in red) located respectively in the upper, middle, and lower parts of the clivus (3D reconstruction by Hélène Coqueugniot).**

In later stages of the disease, the TB infection can spread to other cisterns (e.g., prepontine and ambient cisterns) [52,133–135]; finally, it can reach the ependymal surfaces of the ventricles, the choroid plexuses, and the cerebral convexities [52,128,132,133,135]. In **MMG3-75**, the TB meningeal exudate may have extended towards the interpe-duncular cistern; its presence on top of the upper-middle clivus may have initiated bone resorption resulting in the mild cortical erosion observed (Figs 3–4 and 8c) [28,30,35,40,44,48–51]. Later, not only the cortical layers but also the cancellous bone of the clivus may have been affected by the TB infection; eventually, its granulomatous inflammation may have led to the development of the multiple, very small, well-circumscribed perforating osteolytic lesions in the upper-middle clivus (Figs 3–4) and the three larger, well-circumscribed osteolytic lesions within the upper, middle, and lower parts of the clivus of **MMG3-75** (Figs 8a and 9) [29,30,33,35,39,41–44].

## Corroborative evidence for TB in MMG3–75

Unfortunately, aDNA analyses did not detect *Mycobacterium tuberculosis* DNA in the two hard tissue samples (a tooth and a skull fragment) taken from **MMG3–75**. Consequently, the only corroborative (macromorphological) evidence supporting a diagnosis of TB(M) in this younger adult male comes from the presence of GIs in multiple locations on the inner skull surface. It is important to note that although in the palaeopathological literature, endocranial GIs have been described as specific signs of TBM [99–101,103,104,107], in a recent study by Genchi & colleagues [136], the authors argued against that GIs can be considered as pathognomonic features of clinically active TB(M) disease; instead, they stated that GIs should be interpreted as signs of TB infection. The authors' argument is based on that they found a quite high frequency of cases with GIs in a mediaeval (10th–12th-century-CE) osteoarchaeological series (Pieve di Pava, Italy) [136] that seems to be in conflict with modern disease prevalence data as TBM is a very rare extra-pulmonary form, occurring in less than 1% of all clinically active TB cases today [132,137,138]. Genchi & co-workers [136] state that their hypothesis is supported by 1) the short duration of TBM that is not long enough to allow bony changes (e.g., GIs) to develop on the endocranial

surface and 2) the presence of cases with GIs not only among TB patients but also non-TB individuals from documented skeletal collections (Hamann-Todd Human Osteological Collection [102] and Robert J. Terry Anatomical Skeletal Collection [104,107]). Given that in **MMG3–75** the palaeopathological diagnosis of TBM was established based on the presence of GIs in multiple locations on the inner skull surface, and considering that these lesions constitute the only corroborative evidence indicating that this younger adult male suffered from TB at the time of death, it is inevitable to critically examine Genchi & colleagues' above arguments [136]. A short summary of the critical analysis of the authors' [136] hypothesis can be found below (for a more detailed refutation of Genchi & co-workers' [136] arguments see S1 Text):

1) Genchi & colleagues [136] calculated only the frequency of cases with GIs (and based on this, the frequency of TBM), not the prevalence of TBM, in the examined osteoarchaeological series from mediaeval Pieve di Pava. Then, they compared the estimated frequency of TBM to modern disease prevalence data. However, only the prevalence of TBM in mediaeval Pieve di Pava may be appropriate to be compared to modern disease prevalence data. As it was highlighted by Milner & Boldsen [139], estimating the prevalence of a disease (e.g., TBM) in osteoarchaeological series is no easy task, and is much more complicated than considering only the frequency of a lesion type (e.g., GIs) or the number of diagnosed cases (e.g., number of diagnosed TB(M) cases) in the examined sample;

2) Only disease prevalence data deriving from a representative sample may be appropriate to be used for comparing to modern disease prevalence data but at the current state of research, it is not possible to properly estimate the prevalence of TBM in the once-lived population of mediaeval Pieve di Pava: a) the sample examined by Genchi & co-workers [136] does not seem to be representative of the once-lived population of mediaeval Pieve di Pava (e.g., children under the age of 2 years are missing; and the cemetery is associated with a religious building) [139] and b) although there are some promising palaeoepidemiological studies [e.g., 112] that try to assess the prevalence of TB disease in osteoarchaeological series, further research is needed to develop better methods;

3) By comparing the frequency of cases with GIs (and based on this, the frequency of TBM) in mediaeval Pieve di Pava to modern disease prevalence data and then, stating that GIs cannot be pathognomonic features of clinically active TB(M) disease (as otherwise the prevalence of TBM would be too high in the once-lived population of Pieve di Pava compared to a prevalence of less than 1% of clinically active TB cases today), Genchi & colleagues [136] assume that the prevalence of TB(M) in the past (e.g., mediaeval period) was no different from (or very similar to) its prevalence today. However, the prevalence of TB(M) can be influenced by numerous factors (e.g., nutritional status, age, sex, and genetic factors of the host, and/or genetic factors of the pathogen) [140–141]. It has already been reported in some palaeopathological papers [e.g., 142–144] that before the introduction of the BCG vaccine and anti-TB chemotherapy with antibiotics, the prevalence of TB and its forms (e.g., TBM), and the frequency of some TB-associated bony changes were different from today's trends. On the other hand, our understanding of the global burden of TBM today is limited but it is likely that numerous cases are not diagnosed; thus, the prevalence of TBM, especially in high-TB-burden countries, may be much higher than previously thought [145–146];

4) By questioning the possibility of the development of endocranial bony changes (e.g., GIs) in TBM patients, Genchi & co-workers [136] also assume that the natural history and duration of TB(M) disease in the past (e.g., mediaeval period) were no different from its natural history and duration today. However, like the prevalence of TB(M), these aspects of the disease can be influenced by numerous factors [140–141]. As it has been highlighted by Spekker & colleagues [104] and Madai & co-workers (see Supplementary File 2 in [147]), in the modern medical literature, there are reports of cases (some come from the pre-antibiotic era, some from after the introduction of anti-TB chemotherapy with antibiotics) in which TBM had a protracted course (several months or even years) [e.g., 148–154]. At least in such cases (the prevalence of which is not known neither today nor in the past (e.g., mediaeval period)), the duration of the disease is long enough to allow skeletal lesions (e.g., GIs) to develop on the inner skull surface. On the other hand, the

formation of GIs by pressure atrophy can require several weeks. Thus, even in cases with TBM where there is no initiation of antibiotic therapy and the disease do not have a protracted course but a typical one, the duration of TBM can be long enough to allow endocranial bony changes (e.g., GIs) to develop, especially in consideration that the period between the onset of the signs and symptoms of the disease and the death of the patient, which is preceded by the onset of the formation of tubercles on the meninges, is already several weeks or months (usually two to eight weeks) [155–157];

5) By stating that the presence of cases with GIs not only among TB patients but also non-TB individuals from documented skeletal collections supports that GIs are signs of TB infection rather than clinically active TB(M) disease, Genchi & colleagues [136] seem to overlook that in the two documented skeletal collections they mentioned (Hamann-Todd Human Osteological Collection [102] and Robert J. Terry Anatomical Skeletal Collection [104,107]), individuals were divided into two groups based on what they died of – TB group (those who died of TB) and non-TB group (those who died of non-TB causes) – as there is information about their recorded cause of death only. We do not know the complete medical history of these individuals; thus, we do not know if someone who died of a non-TB cause (e.g., accident, cancer or cardiovascular disease) and was placed into the non-TB group had clinically active TB(M) disease or not (i.e., was a TB(M) patient or not) at the time of death. Based on the above, in the non-TB groups of the aforementioned documented skeletal collections, there may be at least a few TB(M) patients [102,112] and this is why it is not surprising that some of the individuals who died of non-TB causes presented GIs, indicating that they suffered from clinically active TB(M) disease;

6) The Hamann-Todd Human Osteological Collection and the Robert J. Terry Anatomical Skeletal Collection were established when and where many people, possibly everyone, were exposed to TB bacilli [102,112,158]; thus, individuals from the above-mentioned documented skeletal collections are likely to had TB infection – in the TB group, they were not only infected by TB but the infection progressed into clinically active TB disease in all cases, whereas in the non-TB group, they had at least latent TB infection that may have progressed into clinically active TB disease in an unknown number of cases. Based on the above, if Genchi & co-workers' [136] hypothesis would be correct and GIs would be signs of TB infection rather than clinically active TB(M) disease, the frequency of cases with GIs in the non-TB groups should be very similar to their frequency in the TB groups. However, as it has been presented by Spekker & colleagues [104], in the Robert J. Terry Anatomical Skeletal Collection, the frequency of cases with GIs was about 10 times higher in the TB group than in the non-TB group. This difference between the two groups was statistically extremely significant, which confirms that GIs are signs of clinically active TB(M) disease rather than TB infection;

7) If Genchi & co-workers' [136] hypothesis would be correct and it would be the meningeal tubercles developing at the beginning of the TB infection rather than the meningeal tubercles forming during the pathogenesis of TBM that exert pressure on the adjacent endocranial surface for a long time (not weeks but more likely years or decades); and thus, lead to the development of GIs, the frequency of cases with GIs should be much higher not only in the non-TB groups of the Hamann-Todd Human Osteological Collection and Robert J. Terry Anatomical Skeletal Collection, where the majority of the individuals (if not all of them) are likely to had at least latent TB infection, but also in the TB groups, where all individuals are unquestionably to had not only TB infection but clinically active TB disease. However, as it has been presented by Spekker & colleagues [104], even in the TB group of the Robert J. Terry Anatomical Skeletal Collection, GIs were present in less than one-third of the examined cases; and

8) The localisation pattern of GIs on the endocranial surface is not random as we would expect it if Genchi & co-workers' [136] hypothesis would be correct and it would be the meningeal tubercles forming at the beginning of the TB infection after haematogenous dissemination of TB bacilli into the central nervous system that exert pressure on the inner skull surface for a long time; and thus, result in the development of GIs. As it has been presented by Schultz [99], Schultz &

Schmidt-Schultz [103], and Spekker & colleagues [104], instead of being randomly scattered on the endocranial surface, GIs are localised on the skull base and lower lateral skull vault. During the pathogenesis of TBM, the formation of meningeal tubercles presents a very specific pattern (first the disease involves the meninges of the basal areas but in the end, it extends to the meninges covering the cerebral convexities). The localisation pattern of GIs on the endocranial surface follows this specific pattern of tubercle development during the pathogenesis of TBM (see Fig 1 in [104]).

In summary, the currently available scientific evidence is not in favour of Genchi & colleagues' [136] hypothesis that GIs are signs of TB infection (rather than clinically active TB(M) disease) but supports the previous hypothesis [99–101,103,104,107] that GIs are pathognomonic features of TB(M) (rather than TB infection). Therefore, based on the presence of GIs in multiple locations on the endocranial surface of **MMG3–75**, this younger adult male suffered from TBM.

**Differential diagnosis of the clival lesions in MMG3–75**

Although the overall nature and co-occurrence of the clival bony changes observed in **MMG3–75** (mild cortical erosion and multiple, well-circumscribed osteolytic lesions) suggest that TB (specifically TB clival osteomyelitis) could plausibly have been the underlying cause, and the presence of endocranial GIs in multiple locations on the endocranial surface of **MMG3–75** provides corroborative evidence that this younger adult male suffered from TB(M) at the time of death, we cannot rule out the possibility that the clival lesions were associated with another pathological condition that was concurrently present with TB in **MMG3–75**. This consideration makes a detailed differential diagnosis of the clival bony changes indispensable. The most relevant infectious and non-infectious aetiologies that should be considered include clival malignancies (e.g., chordoma, chondrosarcoma or metastases), pyogenic osteomyelitis (e.g., caused by *Pseudomonas* or *Staphylococcus* spp.), and fungal osteomyelitis (e.g., caused by *Aspergillus* or *Mucor* spp.) [24,26,50,53,68,69].

1) Chordomas are relatively rare, malignant bone tumours of the axial skeleton that account for about 2–4% of all primary bone malignancies [159–163]. These slow-growing, locally aggressive, and highly recurrent but rarely metastasising bone tumours originate from embryonic remnants of the primitive notochord [54,58,159–162,164–166]. The skull base is affected in about one-third of all chordoma cases, with the spheno-occipital synchondrosis of the clivus being the most common site of involvement (i.e., clival chordoma) [54,55,58,159,161,165–169]. Clival chordomas are usually centrally located (midline), well-circumscribed, expansile, lobulated soft-tissue masses, which cause extensive lytic bone destruction without evidence of surrounding sclerosis (i.e., well-defined or ill-defined osteolytic lobulated lesions without sclerotic margins) [54,55,58,159,165,167,170–172]. They may present at any age but most occurs in patients between the ages of 20 and 40 years [58,167,170,173–175]; there appears to be no sex difference [170,176–178].

    The biological sex (male) and estimated age at death (30–39 years) of **MMG3–75** do not argue against the diagnosis of clival chordoma, as these malignant bone tumours show no sex predilection and most commonly affect individuals belonging to the 20–40-year age group. In contrast, the osteolytic lesions in **MMG3–75**, although also lacking surrounding sclerosis as typically seen in this tumour type, display a markedly different spatial pattern. They are neither situated along the midline nor confined to the spheno-occipital synchondrosis but are instead scattered across the clivus of this younger adult male in an apparently random manner (Figs 3–4 and 8–9). Furthermore, the three larger osteolytic lesions identified within the clivus of **MMG3–75** are clearly separated from one another and show no evidence of interconnection, thus lacking the lobulated appearance that would be expected in a clival chordoma (Fig 9). Taken together, the combination of the spatial distribution pattern and non-lobulated morphology of the osteolytic lesions observed in this younger adult male is difficult to reconcile with a clival chordoma, which typically arises from the spheno-occipital synchondrosis and presents as a centrally located (midline), well-circumscribed, expansile, lobulated soft-tissue mass. Based on these observations, clival chordoma can be ruled out with high certainty as a diagnostic option in **MMG3–75**;

2) Chondrosarcomas constitute a rare, heterogeneous group of malignant bone tumours of the appendicular and axial skeleton, which make up about 20% of all primary bone malignancies [178–180]. The skull base is involved in about 1% of all chondrosarcoma cases, with a predilection for the synchondroses, especially the petro-clival synchondrosis [179–187]; most of these tumours are located off-midline [159,182,185,188]. Skull-base chondrosarcomas are slow-growing, locally aggressive tumours with a high local recurrence rate and low metastatic potential, which are considered to originate from multipotent mesenchymal cells or from embryonic cartilaginous remnants of the cranium [177,178,186,187,189–194]. They are usually laterally located (off-midline), well-circumscribed, expansile, lobulated soft-tissue masses with a ring-and-arc pattern of chondroid calcification, which cause lytic bone destruction (i.e., ill-defined osteolytic lesions) [159,171,172,177,178,180,186,189]. Skull-base chondrosarcomas can develop at any age but tend to occur in middle-aged adults, and affect females and males equally [58,178,181,187,189,192].

   Although the biological sex (male) and estimated age at death (30–39 years) of **MMG3–75** do not rule out the possibility of a skull-base (clival) chondrosarcoma, he falls slightly below the typical age range (middle-aged adults) for this tumour type. Furthermore, several morphological features, particularly the spatial distribution of the osteolytic lesions in the clivus of this younger adult male, argue against this diagnosis. The well-circumscribed lesions are neither laterally located (off-midline) nor confined to the petro-clival synchondroses, but are instead randomly scattered across the clivus of **MMG3–75** (Figs 3–4 and 8–9), with the three larger osteolytic lesions being spatially separate and showing no signs of interconnection. Although a larger, non-perforating osteolytic lesion is present within the clivus of this younger adult male on each side, adjacent to the petro-clival synchondroses, a skull-base (clival) chondrosarcoma would be expected to arise in the synchondral cartilage and progressively erode the surrounding bone, compromising the cortical layers and producing perforating lesions. In contrast, CBCT images (Fig 8) reveal no evidence of erosion in the bones adjacent to either petro-clival synchondrosis (neither in the laterally adjacent petrous apex nor in the medially adjacent clivus), providing strong evidence that these osteolytic lesions did not develop from the synchondral cartilage inward into the surrounding bone. The observed spatial distribution of the osteolytic lesions detected in **MMG3–75**, together with the intact cortical bone adjacent to the petro-clival synchondroses and the lack of the characteristic ring-and-arc calcification pattern on the CBCT images (Fig 8), is difficult to reconcile with a skull-base (clival) chondrosarcoma, which typically presents as a laterally located (off-midline), well-circumscribed, expansile, lobulated soft-tissue mass with a ring-and-arc pattern of chondroid calcification, mostly arising from the petro-clival synchondrosis. Taken together, these observations strongly suggest that a skull-base (clival) chondrosarcoma is unlikely to account for the mild cortical erosion and multiple, well-circumscribed osteolytic lesions observed in **MMG3–75**;

3) Bone metastases or secondary bone malignancies are quite frequent in adult cancer patients, particularly those with advanced-stage disease [195–197]. From the primary tumour, cancer cells usually enter the skeleton via the bloodstream, but lymphogenous dissemination or contiguous extension can also occur [198–199]. Once in the bone, cancer cells disrupt the normal bone remodelling process; thus, causing an imbalance between bone resorption and bone deposition [200]. Based on whether bone resorption or bone deposition is more dominant, the disease can be classified as 1) (predominantly) osteolytic, 2) (predominantly) osteoblastic, or 3) mixed [172,195,197,199,201,202]; consequently, it exhibits highly variable morphological patterns and radiological appearances [172]. Bone metastases are almost always multiple and primarily affect the bones of the axial skeleton, especially the ribs, vertebrae, and skull [172,195,197,199,203,204]. Rarely, bone metastases can occur at the skull base, in the clivus region [58,205–209]; they usually originate from prostate, kidney, or liver cancers [54,171,205,207,208,210]. Because of the heterogeneous nature of the primary tumours, clival metastases are diverse – they can either be osteolytic or osteoblastic [54,171]. Middle-aged or older individuals are at highest risk of developing clival metastases, with the median age at diagnosis being around 60 years [211–212]; there appears to be a male preponderance [205,211,212].

Although the biological sex (male) of **MMG3–75** is consistent with the slight male predominance reported for clival bone metastases, his estimated age at death (30–39 years) is younger than that of the older individuals typically affected by this pathological condition. Furthermore, beyond the clivus, no similar osteolytic lesions were identified elsewhere in the skull or postcranial skeleton of this younger adult male, whereas secondary bone malignancies usually present as multiple alterations, most often involving different parts of the axial skeleton. Finally, all clival changes in **MMG3–75** are purely lytic, with no evidence of new bone formation. However, purely osteolytic or osteoblastic metastatic lesions are exceedingly rare; in most cases, both osteoclastic resorption and osteoblastic new bone formation are present to some extent, resulting in mixed alterations. Considering the relatively young estimated age at death of **MMG3–75**, the purely lytic nature of the clival lesions and their restriction to the clivus (with no similar alterations detected elsewhere in the axial skeleton), bone metastasis can be regarded as a less likely diagnosis in this younger adult male; and

4) Osteomyelitis is an acute, subacute or chronic inflammatory condition of bone secondary to an infectious process, typically bacterial (e.g., caused by *Staphylococcus* or *Streptococcus* spp.) or less frequently fungal (e.g., caused by *Aspergillus* or *Candida* spp.) [213–223]. It can be limited to the bone but the bone marrow, the periosteum, or the surrounding soft tissues may also become affected by the infection [25,215,218,223,224]. Osteomyelitis can arise subsequent to haematogenous spread of the pathogens from a primary site of infection into the bone [213,215,217,219,220]. It can also occur from contiguous extension of the infection from adjacent structures or from direct traumatic or surgical inoculation of the pathogens into the bone [213,215,217,219,220]. Although any bone of the human skeleton can be targeted by osteomyelitis, the skull is an uncommon localisation [25,213,225,226]; either the skull vault or the skull base can become involved [25]. Skull-base osteomyelitis is usually a direct complication of a bacterial or fungal infection in neighbouring tissues (e.g., otogenic, sinogenic, or odontogenic infections), usually affecting elderly diabetic or immunocompromised patients [25,27,227–233]. The disease can be classified as typical or atypical/central [27,229,231]. The more frequent form, typical skull-base osteomyelitis, is associated with a preceding severe ear infection (e.g., *otitis externa* or *otitis media*), with *Pseudomonas aeruginosa* being the most common causative agent [231,234–237]. It initiates in the temporal bone but can extend to the clivus [231,238]. The much less frequent form, atypical skull-base osteomyelitis, does not have an identifiable otogenic cause [231,235–237]; it is usually secondary to a paranasal sinus infection (e.g., sphenoid sinusitis) and primarily involves the clivus [25,231,236,237]. In pyogenic osteomyelitis, osteolytic lesions are typically ill-defined, often with thick sclerotic margins; reactive new bone formation in the surrounding bone and sequestrum are common [71–73,239,240]. Fungal osteomyelitis can show similar features (e.g., ill-defined osteolytic lesions or marginal sclerosis), but they are less consistent and may be absent [240–241].

The estimated age at death (30–39 years) of **MMG3–75** is not in favour of the diagnosis of pyogenic or fungal skull-base (clival) osteomyelitides, as he is considerably younger than the elderly individuals typically affected by these pathological conditions. Moreover, these osteomyelitides usually arise secondary to ear or paranasal sinus infections, but CBCT images (Fig 8) show that both the middle and inner ears of **MMG3–75** were intact, and the posterior wall of the sphenoid sinus – which became visible due to *post-mortem* damage – was also unaffected (Fig 4b). Although marginal sclerosis, reactive new bone formation, and sequestrum are frequently observed at the osteolytic lesions in such osteomyelitides, especially in pyogenic cases, none of these features were detected in this younger adult male, as demonstrated by the CBCT images (Fig 8). Considering the relatively young estimated age at death of **MMG3–75**, the absence of any evidence for preceding ear (i.e., *otitis media* or *otitis interna*) or paranasal sinus infection (i.e., sphenoid sinusitis), and the lack of marginal sclerosis, reactive new bone formation or sequestrum, it is highly improbable that a pyogenic or fungal skull-base (clival) osteomyelitis was responsible for the development of the mild cortical erosion and multiple, well-circumscribed osteolytic lesions observed in this younger adult male.

## Conclusions

In summary, the younger adult male (**MMG3–75**) from the 16th-century-CE mass grave No. 3 of the Mohács National Memorial Site (Sátorhely, southwestern Hungary) represents a unique case of OATB with regard to the localisation and nature of the observed skeletal lesions. This uniqueness stems from two major aspects:

1) Based on the macromorphological characteristics and co-occurrence of the clival bony changes (mild cortical erosion and multiple, well-circumscribed osteolytic lesions), and after careful consideration of the possible differential diagnoses, the alterations are most consistent with TB clival osteomyelitis. The clivus is an extremely rare site of involvement in TB, with fewer than two dozen of cases reported to date [21,23,24,26–28,50,53,59,61–68]; and

2) Based on the presence of GIs (and ABVIs) in multiple locations on the endocranial surface, **MMG3–75** also suffered from TBM. In the existing modern medical literature, TB clival osteomyelitis associated with meningitis has been mentioned in only a few cases [21,27,28,53,65,67].

Nevertheless, several limitations should be considered when interpreting our findings:

1) Although the pathological bony changes observed in the skull remains of **MMG3–75** correspond to several diagnostic criteria for TB compiled from the palaeopathological literature (summarised in Table 1), indicating that this younger adult male may have suffered from TB with concomitant involvement of the clivus (i.e., TB clival osteomyelitis) and the meninges (i.e., TBM), it was necessary to employ a three-step analytical approach to substantiate this hypothesis.

   First, to demonstrate that the formation of the cranial lesions detected in **MMG3–75** is plausible in the context of TB, we had to reconstruct the possible pathogenetic scenarios that could account for their development. Two pathways were proposed: (1) TB first affected the clivus (i.e., TB clival osteomyelitis) and subsequently extended to the meninges (i.e., TBM) (S2 Fig), or (2) it involved the meninges first (i.e., TBM) and then reached the clivus (i.e., TB clival osteomyelitis) (S3 Fig). It should be noted that although we were able to outline these potential pathogenetic scenarios, illustrating how the cranial bony changes could have developed if our hypothesis is correct (i.e., if TB was indeed the underlying cause in **MMG3–75**), unfortunately, we could not clarify which of these scenarios took place in **MMG3–75**, nor could we establish the primary focus of the TB infection within this younger adult male's body (from where the TB bacilli presumably first reached either the clivus (S2 Fig) or the meninges (S3 Fig) via haematogenous spread).

   Second, to reinforce that TB could indeed plausibly have been the underlying cause of the cranial lesions observed in **MMG3–75**, it was necessary to obtain corroborative evidence that this younger adult male suffered from the disease at the time of death. The presence of GIs in multiple locations on the inner surface of the skull of **MMG3–75** provides such (macromorphological) evidence supporting a diagnosis of TB(M), which may also underlie the formation of the detected endocranial ABVIs. The observation of clival bony changes, potentially consistent with TB clival osteomyelitis, suggests that TB may have contributed not only to the development of the GIs and ABVIs but also to these lesions, and that this younger adult male may have been affected by both disease forms (i.e., TBM and TB clival osteomyelitis) simultaneously. We outlined two plausible pathogenetic scenarios explaining how the TB infection might have spread between the clivus and the meninges (either from the clivus to the meninges or vice versa), and how, in each case, the detected cranial alterations could have formed, with the GIs and ABVIs resulting from TBM and the clival bony changes from TB clival osteomyelitis (S2 Fig and S3 Fig). Nonetheless, a direct causal link between TB and the clival lesions cannot be established, as the presence of GIs in **MMG3–75** provides (macromorphological) evidence only for TBM, which cannot account for the clival bony changes, and not for concurrent TB clival osteomyelitis, which could explain these alterations. Therefore, while TB clival osteomyelitis is indeed a plausible underlying cause of the observed clival lesions, we cannot conclusively assert that it occurred alongside TBM in this younger adult male, nor can we exclude the possibility that another pathological condition concomitantly present with TB in **MMG3–75** contributed to the formation of these alterations.

Third, to substantiate that TB clival osteomyelitis was indeed the most likely primary diagnosis underlying the bony changes detected in the clivus of **MMG3–75**, we had to conduct a detailed differential diagnosis. During this process, both non-infectious (chordoma, chondrosarcoma, and bone metastases) and infectious (pyogenic and fungal osteomyelitides) aetiologies were considered. Taking into account the macromorphological characteristics of the alterations, together with the individual's age at death and sex, we were able to either exclude or consider the aforementioned pathological conditions as less likely to explain the clival lesions observed in **MMG3–75**. Consequently, TB clival osteomyelitis is regarded as the most plausible – and therefore primary – diagnosis in the case of this younger adult male. Unfortunately, it still cannot be ruled out with absolute certainty that another pathological condition, concurrently presenting with TB in **MMG3–75**, contributed to the development of these bony changes, as neither the individual lesions nor their co-occurrence can be regarded as pathognomonic of TB (specifically TB clival osteomyelitis);

2) The only corroborative (macromorphological) evidence supporting a diagnosis of TB(M) in **MMG3-75** comes from the presence of GIs in multiple locations on the inner skull surface, since these endocranial alterations have been described as specific signs of TB(M) in the palaeopathological literature. Unfortunately, attempts to obtain complementary (molecular) evidence for this morphology-based diagnosis were unsuccessful, as the aDNS analysis failed to detect remnants of *Mycobacterium tuberculosis* DNA in the two hard tissue samples (the upper right lateral incisor and a fragment of the left parietal bone) taken from **MMG3-75**.

It is important to emphasise that, although aDNA-based investigations are highly valuable in complementing the findings of traditional morphology-based examinations, they should not be considered as exclusive and indispensable tools for confirming a diagnosis of TB in skeletal human remains. The successful detection of *M. tuberculosis* DNA in prehistoric and historic TB cases depends on multiple factors. In **MMG3–75**, for instance, the aDNA analysis revealed an overall poor preservation of endogenous DNA in both samples, which may account for the negative result. Another possible explanation is that, to avoid irreversible damage to the clival bony changes through invasive sampling and to preserve these morphologically unique and diagnostically significant alterations for potential future investigations, no samples were taken directly from the affected bone area or its immediate surroundings, where the likelihood of detecting *M. tuberculosis* aDNA would presumably have been higher.

Although the endogenous human DNA content recovered from the two hard tissue samples in **MMG3–75** was less than 0.01%, indicating generally poor biomolecular preservation in this younger adult male, this does not necessarily preclude the potential success of alternative molecular diagnostic approaches, such as lipid biomarker analyses. Lipid biomarker-based investigations could provide valuable additional evidence to support the morphology-based diagnosis of TB in skeletal human remains – particularly in cases where endogenous DNA preservation is poor, as in **MMG3–75**. However, like aDNA analyses, these methods also require invasive sampling, preferably from the lesion-affected bone or areas adjacent to it. Targeted sampling of these bone regions would be crucial to demonstrate a direct causal link between the clival alterations and any detected remnants of *Mycobacterium tuberculosis* DNA, and hence between these bony changes and TB itself. Nonetheless, because of the unique nature of the clival lesions observed in **MMG3–75**, we believe that preservation currently outweighs the potential benefits of further invasive molecular investigations. We remain committed, however, to exploring these approaches in the future, once the necessary resources and conditions (e.g., less-destructive sampling techniques) become available; and

3) Finally, it should also be noted that, at present, the contextual interpretation of the case of **MMG3-75** remains limited, as anthropological analyses, including palaeopathological investigations, have so far been completed for only a subset of the individuals buried in mass grave No. 3 of the Mohács National Memorial Site, with the majority of skeletons still awaiting evaluation. Based on preliminary results from the already examined skeletons, several individuals appear to exhibit GIs and other endocranial alterations consistent with TBM. These findings suggest that **MMG3-75** may not have been the only individual affected by TB among those interred in the burial pit. However, to determine the frequency of the disease, and to draw further conclusions from these observations, a comprehensive assessment of the entire

skeletal assemblage will be required. Consequently, a more detailed contextualisation of the case of **MMG3-75** will only be possible upon completion of the ongoing analyses.

Our study contributes to expanding knowledge and understanding of the manifestations and pathogenesis of TB in past human populations by presenting and discussing in detail a unique case of probable TB involvement of the clivus in **MMG3–75**. To the best of our knowledge, **MMG3–75** represents the first reported archaeological case of TB clival osteomyelitis with associated meningitis. It provides an exceptional insight into an extremely rare form of TB, the geographical, temporal, and social distribution of which is still unknown from both prehistoric and historic times, including in mediaeval Hungary from where **MMG3–75** derives. Furthermore, **MMG3–75** not only serves as a reference and aid for the identification and interpretation of similar cases in other human osteoarchaeological series but also offers an invaluable perspective on the disease experience of past individuals who, like this younger adult male, may have suffered from TB clival osteomyelitis.

Although once thought to have been conquered, TB still poses a tremendous public health threat today – largely due to the global emergence and spread of antibiotic-resistant strains. This underlines the urgent need to better understand the disease, including its history, in line with the notion that "*the past informs the present*". Widespread use of antibiotics in the treatment of TB caused certain forms of the disease to vanish or become extremely rare, yet we must now consider the possibility that such pre-antibiotic-era manifestations may reappear or occur more frequently in the future. This is particularly concerning because these disease forms are now almost entirely unfamiliar to clinicians, complicating timely diagnosis and initiation of appropriate treatment, which can negatively impact clinical outcomes in modern cases resembling pre-antibiotic-era manifestations of TB. Although we currently have no information on the prevalence of TB clival osteomyelitis in the past, contemporary knowledge of this disease form is also extremely limited, as the number of reported cases does not exceed two dozen to date. Therefore, beyond its palaeopathological significance, **MMG3–75** also holds relevance for modern medical practice, providing comparative data that may aid in the diagnosis and clinical management of similar modern TB cases, thereby contributing to improved outcomes in such instances. More broadly, our research highlights how archaeological case studies can enhance modern medical knowledge and understanding, helping to bridge the gap between past and present manifestations of TB – one of the world's oldest and still most lethal human infectious diseases.

## Supporting information

**S1 Fig. Damage pattern for human DNA sequences mapping to the hg19 reference genome (table and images by Frank Maixner).**
(XLSX)

**S2 Fig. A possible pathogenetic scenario that could account for the development of the cranial lesions observed in MMG3–75: tuberculosis first affected the clivus (i.e., tuberculous clival osteomyelitis) and subsequently extended to the meninges (i.e., tuberculous meningitis) (image by Marcos De Andrés Montero and Olga Spekker).**
(PDF)

**S3 Fig. A possible pathogenetic scenario that could account for the development of the cranial lesions observed in MMG3–75: tuberculosis involved the meninges first (i.e., tuberculous meningitis) and then reached the clivus (i.e., tuberculous clival osteomyelitis) (image by Marcos De Andrés Montero and Olga Spekker).**
(PDF)

**S1 Text. Detailed refutation of Genchi & colleagues' (2025) hypothesis that endocranial granular impressions are signs of tuberculosis infection rather than clinically active tuberculosis disease.**
(DOCX)

## Acknowledgments

The authors would like to acknowledge the contributions of the directors and other staff members of the Janus Pannonius Museum (Pécs, Hungary) and the Duna-Dráva National Park (Pécs, Hungary). Furthermore, the authors would like to thank the staff of the Király Dental X-ray and CBCT Centre (Szeged, Hungary) for performing the CBCT scanning of **MMG3–75**.

## Author contributions

**Conceptualization:** György Pálfi, Olga Spekker.

**Data curation:** Marcos De Andrés Montero, Frank Maixner, Antal Sklánitz, Árpád Szabó, Gábor Bertók, Hélène Coqueugniot, Olga Spekker.

**Funding acquisition:** Marcos De Andrés Montero, György Pálfi, Olga Spekker.

**Investigation:** Marcos De Andrés Montero, Viktor Vig, Réka Kocsmár, Frank Maixner, Bianca Mari, Alexandra Mussauer, Antal Sklánitz, András Palkó, Ádám Perényi, Árpád Szabó, Tímea Katalin Mai, Gábor Bertók, Olivier Dutour, Hélène Coqueugniot, György Pálfi, Olga Spekker.

**Methodology:** Frank Maixner, Antal Sklánitz, Árpád Szabó, Hélène Coqueugniot, Olga Spekker.

**Project administration:** Olga Spekker.

**Resources:** Albert Zink, Gábor Bertók, Hélène Coqueugniot, György Pálfi.

**Supervision:** Albert Zink, Frank Maixner, András Palkó, Gábor Bertók, Olivier Dutour, György Pálfi, Olga Spekker.

**Visualization:** Marcos De Andrés Montero, Luca Kis, Frank Maixner, Antal Sklánitz, Ádám Perényi, Árpád Szabó, Gábor Bertók, Hélène Coqueugniot, Olga Spekker.

**Writing – original draft:** Marcos De Andrés Montero, Frank Maixner, Bianca Mari, Alexandra Mussauer, Ádám Perényi, Árpád Szabó, Tímea Katalin Mai, Olivier Dutour, Hélène Coqueugniot, Olga Spekker.

**Writing – review & editing:** Marcos De Andrés Montero, Luca Kis, Olga Spekker.

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
