## [Decision Letter · Decision Letter 0]

30 Sep 2025

Dear Dr. Spekker,

Thank you for submitting your manuscript to PLOS ONE. After careful consideration, we feel that it has merit but does not fully meet PLOS ONE’s publication criteria as it currently stands. Therefore, we invite you to submit a revised version of the manuscript that addresses the points raised during the review process.

We look forward to receiving your revised manuscript.

Kind regards,

Mark Spigelman, BSc, MBBS, FRCS

Academic Editor

PLOS ONE

Journal Requirements:

2. In your manuscript, please provide additional information regarding the specimens used in your study. Ensure that you have reported human remain specimen numbers and complete repository information, including museum name and geographic location.

For more information on PLOS One's requirements for paleontology and archeology research, see https://journals.plos.org/plosone/s/submission-guidelines#loc-paleontology-and-archaeology-research .

**Additional Editor Comments:**

PONE-D-25-40158

The first archaeological case of tuberculous clival osteomyelitis with associated meningitis from a 16th-century-CE mass grave from Mohács (southwestern Hungary)

PLOS ONE

Dear Dr Olga Spekker

Your paper is well written and quite comprehensive and together with your excellent and comprehensive bibliography I would like to have it accepted for publication.

I have had 4 reviewers comment on its contents with unusually conflicting results. The paper is worthy of publication but it does require significant changes/alterations.

Reviewer 1: Is harsh in rejecting your paper, but has alerted us to some significant problems but I believe if you approach the comments and provide answers to her criticism, the paper can still warrant publication.

Reviewer 2: Agrees to publication but rises a number of areas whish you should address.

Reviewer 3: Similarly asks for some significant corrections.

Reviewer 4: notes only a small problem with the anatomical description of the skeleton that should be easily corrected.

I would ask you to consider each of the reviewers' views and specifically consider the title to be more general such as "A differential diagnosis of clivial osteomyelitis etc etc" or something similar.

Further the suggestion be more specific about which piece of bone was used for DNA analysis and if it was located close or a part of the actual lesion?. Same for the tooth. This is important, as if he were suffering from active TB and meningitis, it is unlikely he could have been involved in battle as a soldier

The suggestion that you should look at TB-related lipids appears to be an excellent one, as it may make your original suggestion of clivial TB a more acceptable primary diagnosis.

M Spigelman

Reviewers' comments:

Reviewer's Responses to Questions

**Comments to the Author**

1. Is the manuscript technically sound, and do the data support the conclusions?

Reviewer #1: No

Reviewer #2: Yes

Reviewer #3: Partly

Reviewer #4: Yes

2. Has the statistical analysis been performed appropriately and rigorously?

Reviewer #1: N/A

Reviewer #2: N/A

Reviewer #3: N/A

Reviewer #4: N/A

3. Have the authors made all data underlying the findings in their manuscript fully available?

Reviewer #1: Yes

Reviewer #2: Yes

Reviewer #3: No

Reviewer #4: Yes

4. Is the manuscript presented in an intelligible fashion and written in standard English?

Reviewer #1: Yes

Reviewer #2: Yes

Reviewer #3: Yes

Reviewer #4: Yes

Reviewer #1: The paper describes a supposed case of tuberculous osteomyelitis of the clivus in a male individual from a mass grave dated back to the 16th century. The manuscript is well written, but there are crucial criticisms according to which the final diagnosis cannot be supported. The authors start from a pre-defined diagnosis (already anticipated at the beginning of the results!) and continue by proposing it without strong scientific evidence; the differential diagnosis of the lesions to the clivus is oriented to exclude other pathologies, but without strong justification. In summary, as the lesions to the clivus can be caused by many other conditions, and as GIs are not pathognomonic of TB meningitis but only of TB infection (and not disease), the diagnosis of clivus TB osteomyelitis is not strongly supported and lacks empirical evidence.

Major comments

There are no pathognomonic features of skeletal TB in MMG3-75: no characteristic lesions to the vertebrae, ribs, or any other skeletal elements; by admission of the authors, TB osteomyelitis of the clivus is a very rare clinical entity and should be accompanied by lesions of the CVJ, which are absent in the present case. The lesions observed in the skeleton MMG3-75 could be attributed to other pathologies.

There are many other conditions that can cause erosion and lesions on the clivus, as highlighted by the authors in the differential diagnosis, but they exclude them without a strong justification or clear rationale. For example, they exclude skull-base chondrosarcoma because the erosion in MMG3-75 is mild and there are well-circumscribed osteolytic lesions, but they do not consider that this individual could present an initial stage of the tumour. Again, they exclude other conditions based on their clinical rarity, but the TB clivus osteomyelitis is extremely rare as well.

The authors support the diagnosis of TB osteomyelitis of the clivus by linking it to the presence of GIs on the endocranial surface. However, GIs cannot be a manifestation of TB meningitis simply because this is a very rare complication of TB with a very rapid outcome, of which an individual would not live long enough for skeletal manifestations to develop. In this case, the authors affirm that the lesions on the clivus developed following TB infection from the meninges; again, this is not possible as meningitis has a rapid clinical outcome, and the patient should have died before developing the erosion and lesions to the clivus, a process that requires a much longer time. In the paper of Genchi et al. (2025), GIs were found in half of the individuals examined from the Medieval population of Pava, and this strongly demonstrates that they are pathognomonic of TB infection and not TB meningitis; in fact, it is unreasonable to hypothesize that half of the population died from this very rare complication of the central nervous system. Considering that today the prevalence of TB meningitis is 1-5%, and that in the Medieval population of Pava GIs affected half of the skeletal assemblage, even admitting that in the past the prevalence of TB meningitis could have been higher than today, it is absolutely unlikely that the disease reached a prevalence of 50%. The pathophysiology of the human body cannot be changed during a few hundred years! Another point is that cases of protracted course of TB meningitis are reported in the modern clinical literature, but are absolutely exceptional; also, in this case, the authors cannot base their argumentation on rare exceptions, and again if TB meningitis is rapidly fatal today there are no reasons to suppose than in the past the course of this condition was longer enough to leave bone changes. Therefore, to support the hypothesis that GIs are a manifestation of TB meningitis is simply illogical. On the other hand, in the work of Spekker et al., 2020 carried out on the Robert J. Terry Anatomical Skeletal Collection, the prevalence of GIs was observed on individuals with a diagnosis of TB, not TB meningitis; this is a fundamental difference. Additionally, nor in that work nor in previous works reporting GIs as related to TB meningitis (Schultz, 1999; Schultz, 2001; Schultz, 2003; Schultz and Schmidt-Schultz, 2015) it was discussed on a clinical and pathophysiological basis the reason why GIs were supposed as a manifestation of TB meningitis.

Lastly, the final diagnosis of TB is not supported by either molecular analysis or macroscopic assessment of the postcranial bones.

In conclusion, as the lesions to the clivus can be caused by other diseases, and in the case of MMG3-75 there is no strong support that they are caused by TB, as well as the fact that GIs are not pathognomonic of TB meningitis, the diagnosis proposed in this paper is too weak to be sustained.

Minor comments

Page 8, table 1: “secondary hypertrophic osteoarthropathy”, not “hypertrophic pulmonary osteopathy”.

Reviewer #2: First of all, I would like to congratulate the authors for their unique case and for the thorough research carried out.

The paleopathological alterations observed in MMG3-75 are indeed intriguing representation of tuberculosis infection, which has been unseen or remained unnoticed in medieval bioarchaeological research until now.

The findings are well documented, the descriptions are clear, detailed, and precise. The authors did not restrict their research to macromorpholigcal observation solely, but organized and carried out a complex interdisciplinary analysis. While such approaches are getting more and more available, they are unfortunately still not considered standard practice. The evidences are sound and steady, enriched with a rather detailed differential diagnosis.

Despite the complexity and the highly professional manner which are attributing the article, I would have some recommendation for improvement to be considered.

1.) Regarding Figure 1b and Figure 1c, they are good complementary figures. However, if MMG3-75 is not fully covered on the orthophoto, it would be helpful in the positioning if the remains are highlighted or at least marked with an arrow.

2.) The description of the possible scenarios regarding pathogenesis (line 332–360) is detailed and well formulated. However, for readers with different scientific expertise it could be difficult to follow. I would suggest adding a schematic figure summarizing the process either with small pictures or boxes containing one- or two- word prompts for each step.

3.) Although, the section provided for differential diagnosis (line 436–510) is thorough, scientifically accurate, and provides further assurance for the results, it does not highlight the differences and similarities, complicating the understanding. For example, when the authors disputing chordomas (line 444 – 457) — “Chordomas are relatively rare, malignant bone tumours of the axial skeleton that account for about 2–4% of all primary bone malignancies [149-153]. These slow-growing, locally aggressive, and highly recurrent but rarely metastasising bone tumours originate from embryonic remnants of the primitive notochord [54,58,149-152,154-156]. The skull base is affected in about one-third of all chordoma cases, with the spheno-occipital synchondrosis of the clivus being the most common site of involvement (i.e., clival chordoma) [54,55,58,149,151,155-159]. Clival chordomas are usually centrally located (midline), well circumscribed, expansile, lobulated soft-tissue masses, which cause extensive lytic bone destruction without evidence of surrounding sclerosis [54-55,58,149,155,157,160-161]. They may present at any age but most occurs in patients between the ages of 20 and 40 years [58,157,160,162-164]; there appears to be no sex difference [160,165-167]. Based on the above, clival chordoma seems to be less likely to be responsible for the development of the mild cortical erosion and the multiple, well-circumscribed osteolytic lesions observed at the clivus of MMG3-75.” — it is unclear without remembering the detailed description of the lesions which points are making the pathological phenomena relevant and which points are making it excluded from the consideration. Giving a clear comparison in the text or in the form of a table with the key similarities and differences would be beneficial.

In addition to the above, I would like to take the chance to address a questions to the authors.

Due to the poor preservation of endogenous DNA, the palaeogenetic analysis did not provide further evidence of probable tuberculosis infection. Have the authors considered, or are they considering, to collect further evidence based on alternative methods? Although lipid biomarker-based methods are not as commonly applied, they are well-established and provide different approaches for mycolic acid-based diagnosis (Hershovitz et al., 2008; Szewczyk et al., 2013) and mycocerosic acid-based diagnosis (Redman et al., 2009; Váradi et al., 2021).

Finally, I would like to congratulate the authors on this high-quality work. I hope that further paleopathological analyses describing a general picture of the individuals buried in the mass graves of The Mohács National Memorial Site will be also presented in future. This paper has the potential to revive interest in macromorphological and interdisciplinary paleopathological studies, especially those focusing on tuberculous meningitis and osteomyelitis. The importance of this contribution is increasing the knowledge of the evolution of human tuberculosis over the ages, and could be published as it stands.

References:

Hershkovitz I, Donoghue HD, Minnikin DE, Besra GS, Lee OY-C, Gernaey, AM, Galili E, Eshed V, Greenblatt CL, Lemma E, Bar-Gal GK, Spigelman M. (2008) Detection and molecular characterization of 9000-year-old Mycobacterium tuberculosis from a Neolithic settlement in the Eastern Mediterranean. PLOS ONE. 3(10): e3426. DOI: 10.1371/journal.pone.0003426.

Marseille, France & University of Szeged, Faculty of Science, Szeged, Hungary.

Redman JE, Shaw MJ, Mallet AI, Santos AL, Roberts CA, Gernaey AM, Minnikin DE. (2009) Mycocerosic acid biomarkers for the diagnosis of tuberculosis in the Coimbra skeletal collection. Tuberculosis. 89(4): 267-277. DOI: 10.1016/j.tube.2009.04.001.

Szewczyk R, Kowalsky K, Janiszewska-Drobinska B, Druszczyńska M. (2013) Rapid method for Mycobacterium tuberculosis identification using electrospray ionization tandem mass spectrometry analysis of mycolic acids. Diagnostic Microbiology and Infectious Disease. 76: 298–305. DOI: 10.1016/j.diagmicrobio.2013.03.025.

Váradi OA, Rakk D, Spekker O, Terhes G, Urbán E, Berthon W, Pap I, Szikossy I, Maixner F, Zink A, Vágvölgyi Cs, Donoghue HD, Minnikin DE, Szekeres A, Pálfi Gy. (2021) Verification of tuberculosis infection among Vác mummies (18th century CE, Hungary) based on lipid biomarker profiling with a new HPLC-HESI-MS approach. Tuberculosis. 126:102037. DOI: 10.1016/j.tube.2020.102037.

Reviewer #3: The reviewed manuscript is an interesting proposal of a rare location of a bone lesion related to tuberculous meningitis, not described so far. However, in my opinion, several issues need to be addressed before it can be published. Some of them are major problems concerning the differential diagnosis.

The INTRODUCTION is well written, including first a general semblance of tuberculosis at a global level nowadays, including at the end the mention of tuberculous lesion in the clivus as an extremely rare location. However, immediately after that, the aim of the paper is presented, without any clear mention about the relevance of the research or the paleopathological problem that could be solved or answered with the evidence offered by this paper. Thus, in my opinion, a major gap is detected between lines 112 and 113 of the Introduction, where the problems that this paper addresses need to be detailed.

In the MATERIAL AND METHODS, under the Material subtitle, there is not a detailed list (or at least a description) of the present bones in this particular skeleton, nor a description of the preservation of the skeleton, which is extremely necessary for a critical reading about the supported evidence, or the lack of postcranial lesions related to tuberculosis that the authors mention above in the manuscripts.

Also in this section, a map with the location of the archaeological site is included in Figure 1, but it could also include a location at the country or continental level that shows a more visual location to the reader.

In line 170 it is mentioned that the skeleton “is in his 30s at the time of death”, which is very ambiguous. It is necessary here to offer a wider range of estimated ages. Moreover, it is necessary to mention the methods employed for age and sex estimation, instead of just mentioning that the authors used the “methods that are among the standards of practice in the field of bioarchaeology”, as it is written in line 168.

A minor suggestion is to include a new paragraph at the end of line 170, where the paleopathological methods are detailed.

In line 203 the authors mentioned that “tooth and skull fragments” were selected for DNA analysis, but it is also important to mention which tooth and fragments were selected, their preservation and why they were selected.

The RESULTS section begins on line 218, with a statement (i.e., “the skull of MG3-75 showed numerous bony changes that can be attributed to TB”) that has to be the result of the differential diagnosis and not the starting point of the description of the skeletal evidence. Instead, just a clear description of the bone lesions needs to be detailed, as the authors did with the endocranial lesions.

However, at the end of this first paragraph of the Results, there are no mentions of other skeletal lesions, which are described in the list of Table 1 (i.e., spinal and extraspinal bone lesions possibly related to tuberculosis). Thus, it needs to be clarified if any other bone changes were (or were not) recorded in the postcranial skeletal parts.

In the DISCUSSION AND CONCLUSIONS section, again the authors begin with a statement that has to be the result of the differential diagnosis and not the first option to explain the endocranial lesions described before. This statement says, “The pathological skeletal lesions observed in the skull remains of MMG3-75 suggest that this younger adult male suffered from TB with concomitant involvement of the skull base (clivus).” I notice two main and important issues here.

The first one is the lack of evidence that supports the hypothesis of any relationship between the lesions at the clivus and the other endocranial lesions. This is also mentioned in line 319 as a fact, without a previous differential diagnosis. A differential diagnosis of other possible causes that could have affected this particular portion of the skull is just at the very end of this section. No co-occurrence of two (or more) different causes for the recording lesions was explored.

The second issue is the authors assume that TB affected the skeleton of this person based on the proposal that granular impressions are a pathognomonic feature of TB, when in fact it has been discussed by other authors, as the authors discussed further in this section. This is not a problem itself, and I do not have any particular problem with this hypothesis, but this paper does not offer any evidence to support that proposal or for discussing the evidence offered by other authors, as the paper published by Genchi & colleagues does. The discussion then turns to explore if these last authors are or are not right in their hypothesis, which is not the real aim of this paper. Instead, there is not enough space dedicated to the differential diagnosis and to discussing if a clivus lesion could or could not be used as a tool for paleopathological diagnosis of tuberculous meningitis.

In consequence, in my opinion, the discussion is long and disorganized and does not offer a well-sustained answer about the possible relationship of lesions in the clivus and TB. The differential diagnosis at the end of the paper rejects some possible explanations for other causes of these lesions, arguing that they are very rare, but also that the clivus lesions in TB are. Therefore, the evidence offered is not quite convincing.

I suggest also including a separate section of CONCLUSIONS, where the authors also include a list of possible limitations and the relevance of their study.

I hope all these suggestions will be helpful to the authors to improve their research.

Reviewer #4: The manuscript The first archaeological case of tuberculous clival osteomyelitis with associated meningitis from a 16th-century-CE mass grave from Mohács (southwestern Hungary is worth to be published in PLOS ONE. The overall impression is that this study is written in detail and comprehensively. The authors used all known and acknowledged diagnostic criteria for tuberculous clival osteomyelitis and tuberculous meningitis, additionally illustrated with pictures and detailed explanations. The strength of this paper is in listed and explained differential diagnoses, in consulted clinical studies on the basis of which it was possible to reconstruct the first case of tuberculous clival osteomyelitis from archaeological context but also in impressive reference list. Special weight and importance of the work is given by Detailed refutation of Genchi & colleagues’ (2025) hypothesis that endocranial granular impressions are signs of tuberculosis infection rather than clinically active tuberculosis disease in Supporting Information where authors in eight points strengthen the previous hypothesis that GIs are pathognomonic features of TB(M) (rather than TB infection).

This is a useful work and I believe that it can add to the body of literature on the subject of tuberculosis in the ancient populations in general but more specifically in manifestations of tuberculosis of the skull.

There are some minor suggestions that can be added to the manuscript but it is up to authors to accept it or not because they don’t have an extra relevance to the topic of this paper:

In one part of the manuscript was written that “anthropological analysis was carried out on the fairly preserved and almost complete skeleton” and in the other that “It should be noted that the taphonomic damage, as well as the missing elements of the skeleton of MMG3-75 significantly hampered the macromorphological observations”.

Maybe it would be useful to provide an overview of which parts of the skeleton have been preserved, either descriptively or schematically.

Since the skeleton of MMG3-75 derives from a mass grave that contained approximately 320 individuals who were the victims of the Battle of Mohács it would be interesting to mention (although that is not the topic of this paper) whether this male exhibited perimortem trauma.

It would also be interesting to state whether other individuals with signs of tuberculosis were found during the analysis of other skeletons from the mass grave.

**Do you want your identity to be public for this peer review?**  For information about this choice, including consent withdrawal, please see our Privacy Policy

Reviewer #1: No

Reviewer #2: No

Reviewer #3: No

Reviewer #4: No

---

## [Author Response · Author response to Decision Letter 1]

16 Dec 2025

Dear Dr. Spigelman,

We are very thankful for your and the reviewers’ insightful and constructive comments regarding our manuscript entitled “Insights into the pathogenesis and differential diagnosis of clival lesions in an individual from a 16th-century-CE mass grave at Mohács (southwestern Hungary)” that was submitted to PLOS ONE (manuscript ID: PONE-D-25-40158). We are sure that you and the reviewers helped us to improve the quality of our manuscript. The main text and a figure file have been modified following the suggestions, and a new figure and two supplementary figures have been created and added. The revised and new files have been uploaded to the submission site of PLOS ONE.

Responses to the Academic Editor’s comments:

1) “Your paper is well written and quite comprehensive and together with your excellent and comprehensive bibliography I would like to have it accepted for publication.

I have had 4 reviewers comment on its contents with unusually conflicting results. The paper is worthy of publication but it does require significant changes/alterations.”.

Thank you very much for your kind and encouraging words regarding our manuscript. We are truly grateful for your positive evaluation of our work and for highlighting the comprehensiveness of both the manuscript and its bibliography.

We also appreciate the reviewers’ thoughtful feedback. While we recognise that some of the comments reflect differing perspectives, we fully understand that significant revisions are necessary to meet the standards required for publication. During the revision process, we have been committed to carefully addressing all the points raised and to improving the manuscript accordingly.

2) “Reviewer 1: Is harsh in rejecting your paper but has alerted us to some significant problems but I believe if you approach the comments and provide answers to her criticism, the paper can still warrant publication. Reviewer 2: Agrees to publication but rises a number of areas which you should address. Reviewer 3: Similarly asks for some significant corrections. Reviewer 4: notes only a small problem with the anatomical description of the skeleton that should be easily corrected. I would ask you to consider each of the reviewers' views and specifically consider the title to be more general such as "A differential diagnosis of clival osteomyelitis etc etc" or something similar.”.

We have carefully considered all four reviewers’ comments and suggestions, and have done our best to address every concern raised. In the revised version of our manuscript, we have provided detailed responses to each point and made the corresponding changes to the text.

Following your suggestion, we have also revised the title of our manuscript (“The first archaeological case of tuberculous clival osteomyelitis with associated meningitis from a 16th-century-CE mass grave from Mohács (southwestern Hungary)”) to be more general. The new title of our manuscript is: “Insights into the pathogenesis and differential diagnosis of clival lesions in an individual from a 16th-century-CE mass grave at Mohács (southwestern Hungary)”.

We hope that these revisions adequately address the concerns raised and improve the quality of our manuscript.

3) “Further the suggestion be more specific about which piece of bone was used for DNA analysis and if it was located close or a part of the actual lesion? Same for the tooth. This is important, as if he were suffering from active TB and meningitis, it is unlikely he could have been involved in battle as a soldier.”

a) Thank you very much for this important and insightful comment. In the revised version of our manuscript (lines 235–237), we have clarified which specific skeletal elements were used for the aDNA analysis in the Palaeogenetic analysis subsection of the Methods section (we sampled the upper right lateral incisor and a small fragment of the left parietal bone):

“Hard tissue samples (the upper right lateral incisor and a fragment of the left parietal bone) of MMG3-75 were subjected to DNA-based analysis as described by Jäger & colleagues [118].”.

Importantly, the skull sample was not taken from the area of the clival lesions or their immediate surroundings. This was a deliberate decision as we aimed to avoid damaging the pathological bony changes through invasive sampling. Our intention was to preserve the lesions, considering their importance for potential future investigations.

As a comparative example, we would like to refer to a previously studied case of leprosy (SG38) from our skeletal collection, in which a sampling for aDNA and lipid biomarker analyses was conducted directly from the rhinomaxillary region exhibiting pathological (leprosy-related) lesions. Unfortunately, the bony changes were completely destroyed during this previous sampling process, rendering any subsequent re-examination impossible (Tihanyi & colleagues, 2024). We sought to avoid a similar outcome in the case presented in our current manuscript (MMG3-75).

b) We would also like to note that, although the five known burial pits at the Mohács National Memorial Site, including mass grave No. 3, are associated with the aftermath of the Battle of Mohács (1526), at this stage of research, the identities of the individuals buried in these pits are unknown and any interpretation must be considered hypothetical. Over time, three main hypotheses have emerged (Papp, 1961; K. Zoffmann, 1982; Bertók & co-workers, 2022, 2023; Varga, 2023; Drusza, 2024; Viczián & Szeberényi, 2024):

(1) The first hypothesis proposes that the individuals buried in mass grave No. 3 – and, more generally, those interred in the five known mass graves at the Mohács National Memorial Site – were soldiers who fought in the main engagement of the Battle of Mohács and were killed either during the combat itself or while attempting to retreat. This theory has already been questioned. Although the exact location of the battle is currently unknown, all locations proposed to date lie several hundred metres, or even several kilometres, away from the five known mass graves, raising the question of why the Ottoman forces in charge of the burial would have transported the bodies of fallen Hungarian soldiers themselves, or arranged for them to be transported by others, over such a long distance. Nevertheless, the hypothesis cannot be entirely dismissed, as research into the precise location of the combat zone is still ongoing;

(2) The second hypothesis posits that these individuals were members of the Hungarian camp – soldiers guarding it and/or civilians (such as support personnel serving there and/or family members of soldiers) – who may have been killed when the camp was overrun by the Ottoman forces during the battle. This interpretation is supported by recent findings indicating that the location of the five known mass graves can be associated with one of the Hungarian camps established prior to the battle. It is possible that the burial pits were originally defensive ditches surrounding the camp, which were subsequently used by the Ottomans for the interment of the enemy dead; and

(3) The third hypothesis assumes that these individuals were captives (soldiers and/or civilians taken not only during the battle itself, but also in raids preceding or following it), who were systematically executed after the Battle of Mohács; the executions likely took place in the vicinity of the overrun Hungarian camp. This interpretation is supported by both Christian and Ottoman contemporary sources, which mention that Sultan Süleyman did not permit captives to be retained, and that during the victory divan held a few days after the battle (31 August 1526), he ordered the execution of 1,500–2,000 of them. Recent findings estimating that between 1,500–1,700 individuals were buried in the five known mass graves at the Mohács National Memorial Site correspond well with the numbers of executed captives reported in these written sources, thereby strengthening this hypothesis.

It is also possible that those interred in these five burial pits do not represent a single, homogeneous group, but rather a mixture of people executed after the battle, individuals killed when the Hungarian camp was overrun, and others who perished nearby during the fighting or retreat. In other words, the composition of the five known mass graves may not reflect an “either-or” scenario, but rather a combination of these different circumstances.

The taphonomic processes that have taken place over the 500 years since the burial of the individuals have resulted in a variably, but overall poorly preserved and partially commingled state of the more than 300 skeletons interred in mass grave No. 3. This, together with the absence of associated grave goods, greatly hinders efforts to determine the identity of these individuals and the circumstances of their deaths (a situation that also applies to the other four known mass graves at the Mohács National Memorial Site). Observations on other skeletons from mass grave No. 3 – such as evidence of advanced-stage syphilis in at least one individual, signs of tuberculous meningitis in several individuals, and the presence of skeletal elements of adolescents (12–14 years of age based on unfused epiphyses) and possible adult female(s) – further raise the question of whether all individuals interred here were directly involved in the Battle of Mohács.

Consequently, at the current stage of research, none of the aforementioned hypotheses regarding the identity of the individuals buried in mass grave No. 3 (or in the other four known mass graves at the Mohács National Memorial Site) or the circumstances of their deaths can be definitively confirmed or ruled out, and other, less-explored possibilities – such as these individuals being inhabitants of nearby settlements who were killed while assisting retreating Hungarian troops or during Ottoman raids – also remain plausible.

Based on the above, a multitude of possibilities must be considered, and the hypothesis that MMG3-75 was directly involved in the Battle of Mohács represents only one among them, so it remains unclear whether they were a soldier fighting in the battle, a member of the Hungarian camp or a civilian from a nearby settlement, etc. Likewise, the exact circumstances of MMG3-75’s death remain uncertain – whether they perished in the battle, died while retreating or were executed afterwards, etc. To clarify this point in the manuscript, we have inserted two sentences immediately following the relevant part of the Materials section (lines 150–156 in the revised version of our manuscript):

“It is important to note that, to date, several hypotheses have been proposed regarding the identity of the individuals buried in the five known mass graves at the Mohács National Memorial Site, suggesting that they may have been soldiers, members of the Hungarian camp or civilians from nearby settlements [77,79]. Ongoing research seeks to reassess these hypotheses and to shed further light on both the identities of these individuals and the circumstances of their death – whether they perished in the battle, died while retreating or were executed afterwards.”.

In addition, it should be emphasised that the Hungarian (Christian) army at the Battle of Mohács was a heterogeneous and socially diverse force, reflecting the feudal levy system and the extraordinary mobilisation pressures of the time (Nagy-L, 2024). It included the king’s personal retinues, banner units raised by magnates and lesser nobility, peasants and other dependents under their authority, as well as foreign mercenary contingents recruited from German, Czech, and other regions. The composition of the army thus ranged from professional or noble soldiers to conscripted civilians and peasants, with considerable variation in equipment, training, and battlefield role among these groups.

From the summer of 1526, the Hungarian crown increasingly resorted to desperate measures to summon troops against the advancing Ottoman forces (Kovács, 2023; Nemes, 2024). Royal orders mandated, in stages, that nobles provide men from among their peasants: initially every fifth, then every second, and ultimately all able-bodied subjects – including the sick, infirm, and even monks – were required to join the army. These summons were accompanied by the ceremonial display of the “bloody sword” (an ancient signal of military levy), which communicated the urgency of the measure and the severe consequences for noncompliance, including accusations of treason, confiscation of property, and execution.

Taken together, this suggests that even individuals with serious health conditions could have been present in the Hungarian forces, either as part of formal military units (professional soldiers or members of a feudal banner) or as conscripted levies (civilians or peasants). Accordingly, MMG3-75 cannot definitively be interpreted as a professional soldier; this younger adult male could have belonged to any of these groups and, despite severe illness, been present within the army.

While it remains uncertain whether MMG3-75 was actually present as a soldier during the Battle of Mohács, even if he was, the severe pathological condition we infer (i.e., tuberculous clival osteomyelitis with associated meningitis) would not necessarily have prevented him from taking an active role either in the main engagement of the battle or in the defence of the camp. Patients with tuberculous clival osteomyelitis, sometimes complicated by tuberculous meningitis, are typically young adults who develop seemingly mild or nonspecific signs and symptoms such as headache, fever, vomiting or weight loss over a period of several months (usually 1–3, but sometimes up to 6–8 months) (Sagar & colleagues, 2018; Flynn & co-workers, 2022; Iyer & colleagues, 2022; Karim & co-workers, 2025). Despite this apparently benign clinical picture at presentation, radiological investigations often reveal extensive and severe intracranial involvement. Therefore, while MMG3-75’s illness may have affected his physical capacity, it cannot be taken as definitive evidence that he was unable to participate in the army, especially considering that, at the time of conscription or even during the battle itself, he may have still been in the relatively benign phase of his disease, and thus would not necessarily have appeared, or felt, severely ill. All of this, however, remains entirely hypothetical.

We hope this clarification resolves any ambiguity regarding the context of the burial and the identity of MMG3-75.

REFERENCES:

Bertók G, Neményi R, Pálfi G, Simon B. 2022. A mohácsi III. számú tömegsír új kutatása. Magyar Régészet 11(1): 44–53. DOI: 10.36245/mr.2022.1.2

Bertók G, Neményi R, Simon B. Az 1526. évi mohácsi csatához köthető tömegsírok felfedezésének története és a III. számú tömegsír új feltárásának eredményei. [The history of the discovery of the mass graves associated with the Battle of Mohács in 1526 and the achievements of the new excavation of mass grave No. 3.]. In: Varga S, editor. Elsüllyedt Mohács. Újabb Tanulmányok a Mohácsi Csatával Kapcsolatos Kutatások Eredményeiből. [Sunken Mohács. New Studies Using the Research Achievements Relating to the Battle of Mohács.]. Budapest, Hungary: Martin Opitz Kiadó; 2023. pp. 9–19.

Drusza T. A mohácsi csata rekonstrukciója. Új adatok – új megközelítés – új hipotézis. [Reconstruction of the Battle of Mohács. New data – new approach – new hypothesis.]. In: Varga S, editor. Temetetlen Mohács. Az 1526. és az 1687. évi csata új kutatási eredményei. [Unburied Mohács. New Research Results on the Battles of 1526 and 1687.]. Budapest, Hungary: Martin Opitz Kiadó; 2024. pp. 463–502.

Flynn WP, Ntuli Y, Zhang H, Tiberi S. 2021. A case of clival tuberculosis and associated meningitis. Journal of Clinical Tuberculosis and Other Mycobacterial Diseases 25: 100273. DOI: 10.1016/j.jctube.2021.100273

Iyer AS, Patil PV, Pandey D, Kute BS, Shetty BB. 2022. Tubercular skull base osteomyelitis – A case report. IDCases 27: e01360. DOI: 10.1016/j.idcr.2021.e01360

Karim S, Kurup B, Solomon R, Shobhavat L, Gandhi D, Verma S. 2025. Nasopharyngeal mass with skull base invasion due to tuberculosis. Indian Journal of Pediatrics 92(10): 1129

---

## [Editor Report · Decision Letter 1]

28 Dec 2025

Insights into the pathogenesis and differential diagnosis of clival lesions in an individual from a 16th-century-CE mass grave at Mohács (southwestern Hungary)

PONE-D-25-40158R1

Dear Dr. O Spekker

We’re pleased to inform you that your manuscript has been judged scientifically suitable for publication and will be formally accepted for publication once it meets all outstanding technical requirements.

Kind regards,

Mark Spigelman, BSc, MBBS, FRCS(London)

Academic Editor

PLOS One

Additional Editor Comments (optional):

Please use your standard letter of reply at your conveniance once you have considered my above comments.

M Spigelman
---

## [Editor Report · Acceptance letter]

PONE-D-25-40158R1

PLOS One

Dear Dr. Spekker,

I'm pleased to inform you that your manuscript has been deemed suitable for publication in PLOS One. Congratulations! Your manuscript is now being handed over to our production team.

Kind regards,

on behalf of

Dr. Mark Spigelman

Academic Editor

PLOS One